# The sponge effect and carbon emission mitigation potentials of the global cement cycle

Zhi Cao [1], Rupert J. Myers [2,3], Richard C. Lupton [4], Huabo Duan[5], Romain Sacchi[6], Nan Zhou[7], T. Reed Miller [8], Jonathan M. Cullen[9], Quansheng Ge [10] & Gang Liu [1,10 ✉]

Cement plays a dual role in the global carbon cycle like a sponge: its massive production contributes significantly to present-day global anthropogenic $CO_2$ emissions, yet its hydrated products gradually reabsorb substantial amounts of atmospheric $CO_2$ (carbonation) in the future. The role of this sponge effect along the cement cycle (including production, use, and demolition) in carbon emissions mitigation, however, remains hitherto unexplored. Here, we quantify the effects of demand- and supply-side mitigation measures considering this material-energy-emissions-uptake nexus, finding that climate goals would be imperiled if the growth of cement stocks continues. Future reabsorption of $CO_2$ will be significant (~30% of cumulative $CO_2$ emissions from 2015 to 2100), but climate goal compliant net $CO_2$ emissions reduction along the global cement cycle will require both radical technology advancements (e.g., carbon capture and storage) and widespread deployment of material efficiency measures, which go beyond those envisaged in current technology roadmaps.

[1] SDU Life Cycle Engineering, Department of Chemical Engineering, Biotechnology, and Environmental Technology, University of Southern Denmark, 5230 Odense, Denmark. [2] Institute for Materials and Processes, School of Engineering, University of Edinburgh, Edinburgh EH9 3FB, UK. [3] Department of Civil and Environmental Engineering, Imperial College London, London SW7 2AZ, UK. [4] Department of Mechanical Engineering, University of Bath, Bath BA2 7AY, UK. [5] School of Civil Engineering, Shenzhen University, 518060 Shenzhen, China. [6] R&D, Quality and Technical Sales Support, Cementir Holding S.p.A., 9220 Aalborg, Denmark. [7] China Energy Group, Energy Analysis and Environmental Impacts Division, Energy Technologies Area, Lawrence Berkeley National Laboratory, Berkeley, CA, USA. [8] Department of Chemical and Environmental Engineering, Yale University, New Haven, CT 06511, USA. [9] Department of Engineering, University of Cambridge, Trumpington St., Cambridge CB2 1PZ, UK. [10] Institute of Geographical Sciences and Natural Resources Research, Chinese Academy of Sciences, 100101 Beijing, China. ✉email: gli@kbm.sdu.dk

C ement is an essential ingredient in concrete and mortar, two construction materials used extensively in the built environment[1]. The rapid growth in demand for cement in recent history has positioned the cement industry as one of the largest energy consumers and $CO_2$ emitters[2,3]. In 2014, cement production contributed ~7% (10.7 EJ) of global industrial energy use and 22% (2.2 Gt) of global $CO_2$ emissions from industrial processes[4]. Conversely, cement-related materials like mortar and concrete are significant $CO_2$ sinks[5] due to their ability to react with (absorb) atmospheric $CO_2$, which is particularly significant in the use and end-of-life stages of the cement cycle[6], approximately equivalent to the total $CO_2$ emissions from international maritime transport[7]. We refer to this dual role in emitting and soaking up $CO_2$ along the entire cement cycle (from production, through use, and to end-of-life) as the "sponge effect", and it must be considered in examining long-term decarbonization pathways and identifying carbon management strategies for this material system.

Although the carbonation effect is well known as a deterioration mechanism of concrete, it has relatively recently been recognized as a potentially significant $CO_2$ sink[6]. The scale of historical $CO_2$ absorption occurred along the entire cement cycle has been estimated regionally[6,8] and globally[5], concluding that nearly half of process emissions in cement production from 1930 to 2013 have likely been sequestered by cement-related materials[5]. Understanding the mitigation potential of the sponge effect requires looking to the future, but future scenarios are often either based on cement demand linked to market growth[9,10] or economic indicators[11,12], or limited to a certain life-cycle stage (e.g., end-of-life demolition waste[13]). A proper, holistic understanding of the sponge effect requires not just forecasting cement demand but also a physically consistent accounting of the cement stocks in the built environment, and end-of-life demolition waste, where the carbonation actually occurs, and the cement demand for replacement and expansion of stocks.

If the world follows a development pathway that is consistent with typical patterns observed in several industrialized countries, the global convergence of buildings and infrastructure services in all nations, to the level of these countries, is expected to drive sustained increases in global cement demand to build up the desired in-use stocks[1,14–16]. Simultaneous expansion, demolition, and replacement of cement stocks in the built environment will generate significant amounts of demolition waste once building and infrastructures reach their end-of-life[17,18], as well as construction waste during their construction, both of which have different $CO_2$ absorption characteristics from cement in active use and account for a large part of the lifetime $CO_2$ absorption[6]. The use patterns of cement stocks and their longevity (a lifetime from decades to centuries) create long-term path dependences for both cement demand and demolition waste generation[19–22]. The explicit characterization of cement flows and stocks enables an explicit understanding of the components of the sponge effect and the resulting net $CO_2$ emissions balance along the future cement cycle, which has been missing in previous work.

To understand the role of the sponge effect along the cement cycle in future $CO_2$ emissions mitigation, in this study, we develop a multilayer dynamic material flow analysis (MFA) model that describes the material-energy-emission-uptake nexus in the global cement cycle from 1930 to 2100. Our model integrates three modules (see "Methods"): first, a global dynamic MFA model[1,19] to determine the past, present, and future stocks and flows (e.g., demand and demolition waste generation) of cement-related materials[23]; second, a global cement technology roadmap[9,24] that projects the development of $CO_2$ emissions mitigation measures in cement production; and third, a physicochemical carbonation model[5] that estimates uptake of

atmospheric $CO_2$ by cement-related materials over time. We project that cement carbonation will gradually reabsorb ~30% of $CO_2$ emissions arising from cement production across nine conceived cement stock scenarios, but deep decarbonization of the global cement cycle entails further improvements in material efficiency at the demand side, as well as step changes at the supply side.

## Results

**Global cement cycle in 2014**. Figure 1 illustrates the 2014 global cement cycle and the associated net $CO_2$ emissions balance (see "Methods"). Driven by the expansion and turnover of in-use stocks, 4.2 Gt of cement and 0.2 Gt of cement kiln dust (CKD) were produced in 2014. Cement stocks in 2014 amounts to ~75 Gt in total, nearly equally split between residential, non-residential, and civil engineering sectors with ~25 Gt each. The longevity of cement stocks means that only 0.5 Gt of demolition waste was generated in 2014. The challenges faced in recycling cement-based products lead to nearly all (99.1%) demolition waste being buried in landfills, or as part of backfills and aggregates in road base (see Supplementary Table 1). We calculate that the global cement cycle gave rise to 3.0 Gt of $CO_2$ emissions and 0.6 Gt of $CO_2$ uptake in 2014, offering a net balance of 2.4 Gt $CO_2$ emissions. Of the total $CO_2$ emissions released from cement production and upstream processes in 2014, 58.4% were released from carbonate calcination, 32.9% from fuel combustion, and 8.6% from indirect emissions for electricity generation. Our result indicates that most of the $CO_2$ uptake (~80%) in 2014 occurred in buildings and infrastructures (in-use stocks), with CKD, construction waste, and demolition waste, together, contributing only ~20% to the total $CO_2$ uptake.

**Decarbonization storylines and scenario narratives**. To understand how the cycle depicted in Fig. 1 could develop in the future, we used a top-down stock-flow approach driven by data on cement production, trade, sectoral use, and lifetime[1], to estimate the historical and contemporary cement stocks. We observed that the per capita cement stocks in all ten regions have increased since 1930 (see Supplementary Figs. 1–10). Global average cement stocks per capita reached 10.2 tonnes per capita in 2014, with industrialized and transitioning regions ranging from 12.7 to 23.7 tonnes per capita, developing regions ranging from 2.7 to 7.5 tonnes per capita, and several mature economies approaching 35 tonnes per capita. However, regional cement stocks are not equally distributed across sectors. Post-industrial regions (especially the Commonwealth of Independent States; CIS) typically have higher levels of per capita cement stocks in the civil engineering sector. In contrast, China has a lower level of per capita cement stocks in the civil engineering sector, but a considerably higher level in buildings. We speculated that these variations could be explained by multiple factors, such as the development stage, patterns of urban expansion, architectural specification, as well as availability and choice of construction materials[1]. Earlier studies have shown a saturation phenomenon for per capita in-use stock development of bulk materials, such as iron[25,26] and copper[27] in industrialized countries, but not for aluminum, due to its relatively short history of use[28]. Likewise, the development patterns of per capita cement stocks generally comply with an S-shaped curve, and saturation is evident in several highly developed countries[1]. The saturation of per capita cement stocks implies that the growth rate of buildings and infrastructures in use (where cement stocks reside) will decrease marginally and eventually reach a plateau, as services provided by cement stocks become saturated[17,29–32]. Furthermore, as evidenced in several highly developed economies[1], decreasing trends of per capita

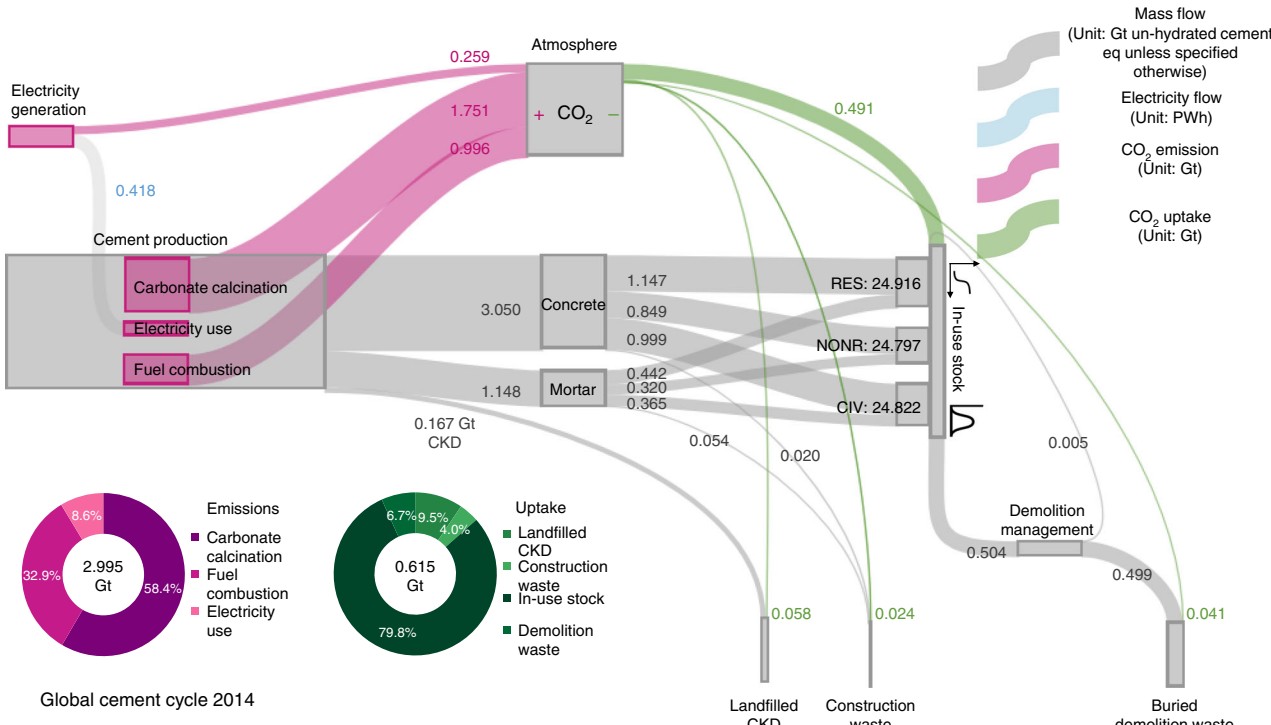

**Fig. 1 Global cement cycle in 2014.** The term cement most commonly refers to hydraulic (chiefly Portland) cement[56]. All stocks and flows of cement-related materials are herein expressed in un-hydrated cement equivalent and excluding inert materials that are used as aggregate in concrete and mortar. Percentages may not add up to 100% due to rounding. RES residential buildings, NONR non-residential buildings, CIV civil engineering, CKD cement kiln dust.

cement stocks have become manifest, reflecting that material efficiency strategies have come to play a significant role in these economies. We therefore envisage three scenario storylines with varying levels of cement stocks similar to the Resource Efficiency-Climate Change Nexus (RECC) scenario modeling framework[33], which is built upon the Shared Socioeconomic Pathway (SSP) scenarios and the Low Energy Demand (LED) scenario[34]; the first scenario storyline (S1–3) is characterized by a low cement stock level, the second scenario storyline (S4–6) by a medium cement stock level, and the third scenario storyline (S7–9) by a high cement stock level. The saturation level of per capita cement stocks is regarded as a tangible indicator for various human needs in mature societies, including shelter, transport networks, factories, offices, as well as commercial, educational, healthcare, and governmental facilities. It is the level of service provided by per capita cement stocks that are expected to saturate, not just the quantity of material involved; the two are linked by the material intensity of the in-use product stocks. Concurrent with the development of cement stocks, demand for cement will slow down, decline, and ultimately stabilize, given that the dynamics of cement stocks, to a large degree, determine the demolition rate and reconstruction rate for cement-related materials, according to the mass-balance principle[21,35].

In light of the observed historical patterns of cement stocks and the essential role of in-use stock dynamics to the cement cycle, we simulate the future cement cycle in ten regions using a stock-driven approach[17] based on the historical patterns of per capita cement stocks identified in our previous work[1], three storyline-consistent target values of per capita cement stocks (i.e., saturation levels), and a moderately growing population obtained from the medium scenario of United Nations World Population Prospects[36]. We deem the level of in-use cement stocks as an explicit physical representation of service provision to society, thereby constructing nine stock-driven scenarios (created from

three saturation levels and three saturation times) to explore the evolution of cement-related materials until 2100 due to the longevity of buildings and infrastructures. Our scenarios build upon three key assumptions: first, per capita cement stocks in the ten regions follow a development path that is consistent with S-shaped curves or inverted S-shaped curves toward a global convergence of per capita cement stocks, and therefore, regions or end-use sectors that have a per capita cement stock below the saturation level will see a continuing growth, while those with a per capita cement stock over the saturation level will see a decline (see Supplementary Fig. 11); second, the formulated pathways of per capita cement stocks do not entail abrupt changes in resulting cement demand, and therefore, the development pathways of per capita cement stocks in a few regions or end-use sectors are adjusted to smoothen the trends in cement demand; third, technological development for optimizing cement use in buildings and infrastructure proceeds, but without fundamental breakthroughs (e.g., new materials that replace cement to a full extent), because cement is a ubiquitous, relatively cheap building material of good workability.

In all of the nine scenarios, we parameterize two boundary conditions, saturation level and saturation time, to reflect the varying patterns of cement stocks and varying levels of future demand-side material efficiency in different regions. By considering a range of saturation levels, we cover both a range of service levels provided by the in-use cement stocks and a range of material efficiencies in their delivery. The saturation time reflects the speed of stock growth (parameterized by the time when the per capita cement stocks reach 98% of the saturation level). Given the regional heterogeneity of socioeconomic and geographic circumstances, we set varying saturation levels and times for different regions to fit the historical development of per capita cement stocks (see Supplementary Table 2). A modified Gompertz model is used for simulating the growth curves of

**Table 1 Supply-side mitigation measures and their implementation.**

| Measure code | Description | Model implementation |
|---|---|---|
| E-M1 | Thermal efficiency | 3.3 GJ t$^{-1}$ Portland cement clinker by 2030 |
| | | 3.2 GJ t$^{-1}$ Portland cement clinker by 2050 |
| | | 2.9 GJ t$^{-1}$ Portland cement clinker by 2100 |
| E-M2 | Electric efficiency | 92 kWh t$^{-1}$ cement by 2050 (applied to NA, LAC, EU, CIS, AF, ME, and CN) |
| E-M3 | Alternative fuel | 20% of alternative fuel by 2030 |
| | | 35% of alternative fuel by 2050 |
| | | 50% of alternative fuel by 2030 (only applied to EU) |
| | | 60% of alternative fuel by 2050 (only applied to EU) |
| E-M4/U-M4 | Clinker substitution | 73% clinker ratio by 2050 (applied to NA, EU, CIS, AF, ME, CN, DAO, and DA) |
| E-M5 | Carbon capture and storage (CCS) | 25% of $CO_2$ emissions from cement production captured in cement plants by 2050 |

*E* $CO_2$ emissions, *U* $CO_2$ uptake, *M* $CO_2$ emissions mitigation, *NA* North America, *LAC* Latin America & Caribbean, *EU* Europe, *CIS* Commonwealth of Independent States, *AF* Africa, *ME* Middle East, *IN* India, *CN* China, *DAO* Developed Asia & Oceania, *DA* Developing Asia.
Note: Full details are delineated in Supplementary Note 3. M4 is coded twice because clinker substitution reduces $CO_2$ emissions of per tonne of cement, as well as $CO_2$ uptake in cement-related materials.

per capita stocks based on assumed saturation levels and times (see Supplementary Note 2.2).

Under the nine stock-driven scenarios, we further characterize the sponge effect and its resulting net $CO_2$ emissions balance for the cement cycle and explore future decarbonization pathways. This includes both demand-side mitigation options to increase material efficiency, reflected in the chosen saturation levels for in-use stocks, and supply-side mitigation options, represented by changes in the $CO_2$ emissions intensity of cement production. We extract five supply-side $CO_2$ emissions mitigation measures from the global cement technology roadmap[4,9] (see Table 1): thermal efficiency (E-M1), electric efficiency (E-M2), alternative fuel (E-M3), clinker substitution (E-M4), and carbon capture and storage (E-M5). Each measure represents an effort beyond what would occur under a no-action scenario; therefore, the remaining $CO_2$ balance is quantified by subtracting the $CO_2$ emissions reduction potentials of the five measures (when they are rolled out simultaneously) from the no-action scenario. The $CO_2$ uptake is explicitly simulated in a physicochemical carbonation model[5] by applying Fick's diffusion law (see "Methods").

**Decarbonization pathways of global cement cycle**. The gradual rise and then saturation of in-use stocks lead to cyclical variations in global cement demand over the next decades (see Supplementary Figs. 12–22), while the global demolition waste generation continues to rise due to the delay between demand and demolition caused by the longevity of in-use cement stocks (see Supplementary Figs. 23–33). Our estimates of cement demand in the year 2050 (4.3–6.7 Gt yr$^{-1}$) are more wide-ranging than those estimated by the International Energy Agency technology roadmap for the global cement industry (4.7–5.1 Gt yr$^{-1}$)[2,9].

Figure 2a shows $CO_2$ emissions under the no-action scenario and the effects of the mitigation measures. In 2050, the no-action $CO_2$ emissions under low-, medium-, and high-saturation levels reach 3.0–3.4 Gt yr$^{-1}$, 3.4–4.0 Gt yr$^{-1}$, and 3.8–4.7 Gt yr$^{-1}$, respectively. In parallel, the $CO_2$ uptake (effects of U-M4 subtracted, the same hereafter) rises to 0.9–1.0 Gt yr$^{-1}$ (low-saturation levels), 1.0–1.1 Gt yr$^{-1}$ (medium-saturation levels), and 1.1–1.3 Gt yr$^{-1}$ (high-saturation levels) by 2050. The no-action $CO_2$ emissions balance (when $CO_2$ uptake is considered) in 2050 increases to 2.1–2.3 Gt yr$^{-1}$ (low-saturation levels), 2.4–2.9 Gt yr$^{-1}$ (medium-saturation levels), and 2.7–3.4 Gt yr$^{-1}$ (high-saturation levels), respectively. By 2100, the balance is at slightly lower levels, ranging from 1.5 Gt yr$^{-1}$ to 3.1 Gt yr$^{-1}$.

By implementing a full portfolio of mitigation measures, $CO_2$ uptake begins to overtake the remaining $CO_2$ emissions from cement production by the late 2090s, bending the net $CO_2$ emissions balance below zero (Fig. 2a). However, in the medium term, the 2050 net $CO_2$ emissions balance of the global cement cycle will reach 1.0–1.2 Gt yr$^{-1}$ (low-saturation levels), 1.2–1.5 Gt yr$^{-1}$ (medium-saturation levels), and 1.4–1.8 Gt yr$^{-1}$ (high-saturation levels), respectively. Of the nine stock-driven scenarios, none generates a trajectory of net $CO_2$ emissions balance that follows, or is below, the 1.5 °C-consistent pathway, meaning excessive $CO_2$ is emitted along all trajectories. If the cement industry is to contribute to the 1.5 °C limit in proportion with other industrial sectors, achieving the $CO_2$ emissions reduction target by employing mitigation measures in the production stage alone is extremely challenging, because net $CO_2$ emissions balance largely hinges on in-use stock dynamics, and concomitant demand and demolition.

Long-term accounting for $CO_2$ uptake along the cement cycle, which could be regarded as passive $CO_2$ sequestration, greatly changes the net $CO_2$ emissions balance of the global cement cycle. Across the stock-driven scenarios, the cumulative $CO_2$ uptake from 2015 to 2100 amounts to 81.1–117.2 Gt (Fig. 2b). These values correspond to roughly 30% of the no-action $CO_2$ emissions arising from the global cement cycle over the same period. All decarbonization pathways are characterized by widespread deployment of CCS technologies (E-M5) in the production stage. From 2015 to 2100, cumulative $CO_2$ emissions mitigated by CCS technologies, which could be regarded as active $CO_2$ sequestration, are 56.7–94.2 Gt, accounting for ~25% of no-action $CO_2$ emissions from cement production (Fig. 2b). We therefore conclude that deep decarbonization of the global cement cycle calls for both passive $CO_2$ sequestration and active $CO_2$ sequestration, but also that these measures are likely not enough to reach the 1.5 °C climate goal— more innovative or drastic approaches are needed.

**Regional disparities of decarbonization potential**. Figure 3 shows that the regional patterns of the sponge effect shift along with the stock dynamics and population trends, resulting in varying cumulative no-action $CO_2$ emissions and mitigation strategies. The population boom and gradual rise of in-use stocks are major factors that drive $CO_2$ emissions in emerging regions, as massive improvements in the provision of shelters and infrastructures in these regions take place. For example, Africa's no-action cumulative $CO_2$ emissions from 2015 to 2100 are 53.9–108.5 Gt. Although China's per capita cement stocks had already peaked in 2014, its cumulative no-action $CO_2$ emissions during 2015–2100 will still reach 61.7–75.6 Gt, due to the shorter lifetimes of in-use cement stocks in China. Meanwhile, the cumulative no-action $CO_2$ emissions that will occur in industrialized regions (NA, EU, CIS, and DAO regions altogether) from 2015 to 2100 are lower, ranging from 22.0 to 47.9 Gt. Compared with other regions, the active $CO_2$

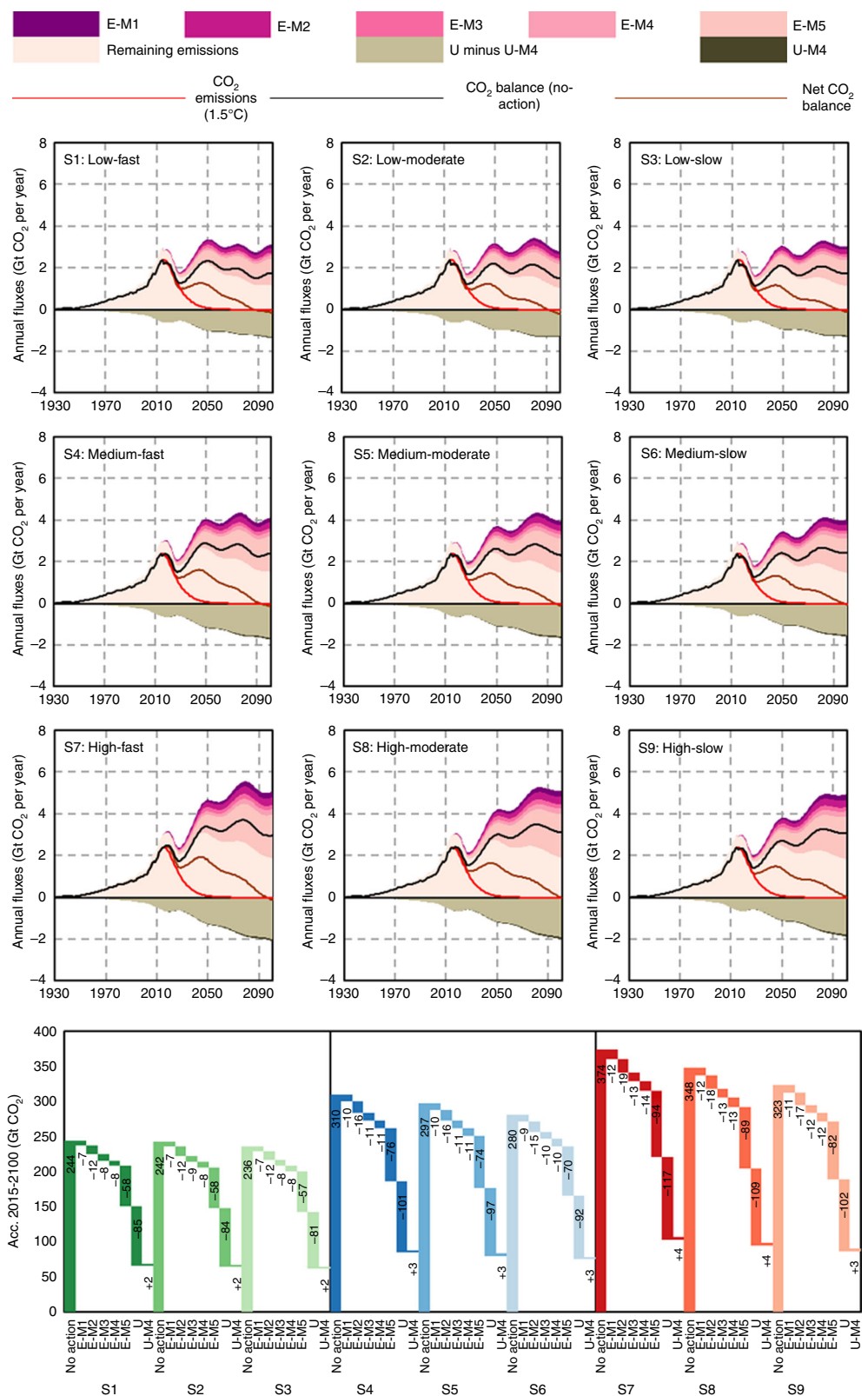

sequestration (E-M5) plays a more dominant role in emerging regions (e.g., ~30% in both Africa and India). This indicates that CCS implementation should take place in the emerging regions where new demand for cement and production facilities increases rapidly. However, CCS is still at the demonstration stage, and their large-scale market deployment is hindered by high estimated costs[37], which is a significant issue for investment constrained emerging economies, suggesting that effective policies, intensified research to reduce CCS costs, and/or international financial support for CCS in cement production are urgently needed. Active $CO_2$ sequestration by CCS can be further utilized (carbon capture and utilization; CCU) as a feedstock to produce chemicals and fuels; however, the development of CCU technologies is still in its infancy and limited to the laboratory scale[37].

**Fig. 2 Decarbonization pathways and supply-side mitigation measures of the global cement cycle across the nine stock dynamic scenarios. a** The no-action $CO_2$ emissions and uptake pathways from 2015 to 2100 coupled with the results of the five supply-side mitigation measures. **b** The 2015–2100 accumulated mitigation potential by the five supply-side mitigation measures and uptake. $CO_2$ emissions (1.5 °C): the red line represents the calculated $CO_2$ emissions pathway that is consistent with the 1.5 °C budgets (a 66.7% probability) in the IPCC's special report (see "Methods"). $CO_2$ balance (no-action): the black line represents the no-action $CO_2$ balance, that is, no-action $CO_2$ emissions minus no-action $CO_2$ uptake. Net $CO_2$ balance: the brown line represents the net $CO_2$ balance when the five supply-side mitigation measures are implemented. U-M4: clinker substitution marginally reduces $CO_2$ uptake in cement-related materials. Acc. accumulated, Low low stock saturation level, Medium medium stock saturation level, High high stock saturation level, Slow slow stock saturation time, Moderate moderate stock saturation time, Fast fast stock saturation time.

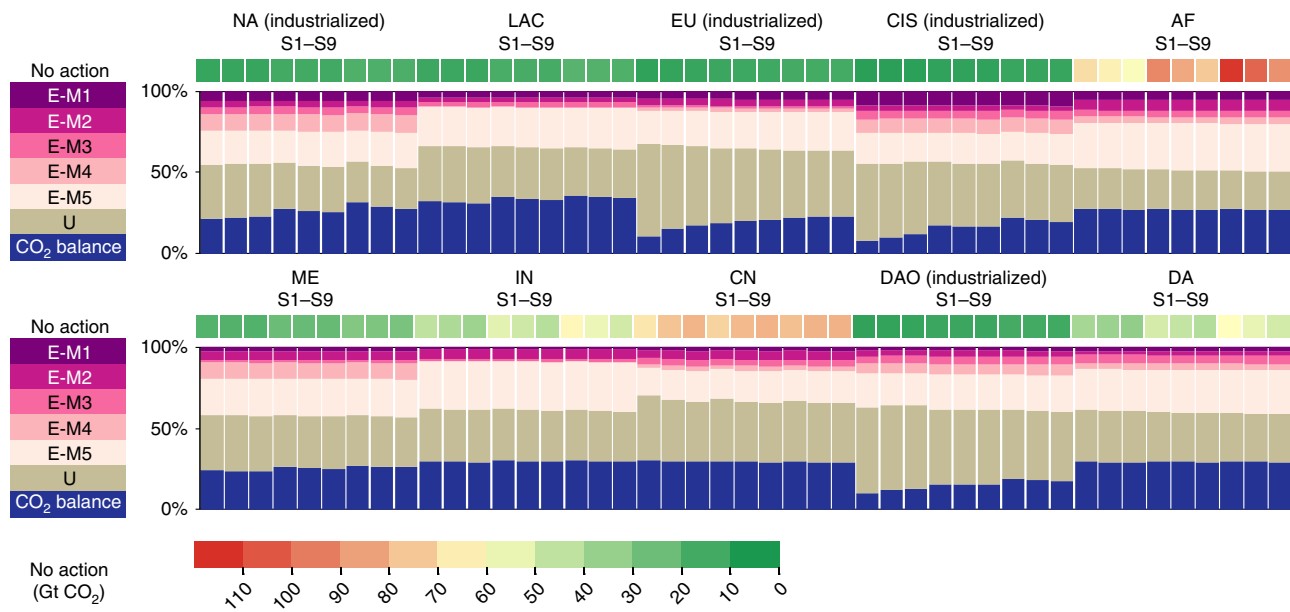

**Fig. 3 Regional patterns of cumulative non-action $CO_2$ emissions (2015–2100) and relative contribution of the five supply-side $CO_2$ emissions mitigation measures, $CO_2$ uptake, and $CO_2$ balance.** The five supply-side measures refer to those listed in Table 1. The red-yellow-green palette represents the 2015–2100 cumulative (no-action) $CO_2$ emissions. The column chart represents the relative contribution of the five supply-side $CO_2$ emissions mitigation measures, $CO_2$ uptake, and the remaining $CO_2$ balance. NA North America, LAC Latin America & Caribbean, EU Europe, CIS Commonwealth of Independent States, AF Africa, ME Middle East, IN India, CN China, DAO Developed Asia & Oceania, DA Developing Asia. S1: low–fast; S2: low–moderate; S3: low–slow; S4: medium–fast; S5: medium–moderate; S6: medium–slow; S7: high–fast; S8: high–moderate; S9: high–slow.

## Discussion

These results clearly demonstrate that any policy or initiative aiming at decarbonizing the cement sector must consider the sponge effect, given that the magnitude of this passive $CO_2$ sequestration is similar to or greater than the active CCS sequestration assumed in the technology roadmap. It represents the interplay between in-use stock dynamics and atmospheric $CO_2$ concentrations, which sets critical boundary conditions for decarbonization in the cement cycle. Again, the analytical results presented in this study should always be interpreted within the formulated scenario narratives. Given this context, the varying saturation levels in our scenario analysis highlight the urgent and precious opportunities to mitigate $CO_2$ emissions in emerging regions where buildings and infrastructures are yet to be constructed. To avoid lock-in effects in the cement cycle, emerging regions should avoid replicating the patterns of cement stocks from developed regions, and instead pursue more ambitious material efficiency strategies[22,38–40] to achieve desired levels of service from smaller cement stocks (see detailed discussion in "Methods"). The range of stock saturation levels across scenarios represents a modest level of effort (~17%; see Supplementary Table 2) given to material efficiency strategies. In contrast, a pilot study in the UK shows that material efficiency strategies could potentially deliver a 50% reduction in cement use[41], indicating that significant $CO_2$ savings remain untapped. The significance of material efficiency

strategies is also examined in a special report led by International Energy Agency, in which a bottom-up analysis of the building sector shows that material efficiency improvements in the buildings sector can reduce ~26% of its annual cement demand in 2060 (see Fig. 25 in ref. [42]). Accounting for both saturation levels and saturation time, in 2060, annual global cement demand sees a 44% decline in the Low–Slow scenario (3.8 Gt yr$^{-1}$) relative to the High–Fast scenario (6.7 Gt yr$^{-1}$).

The atmospheric $CO_2$ exchanges associated with the production, use, and demolition of cement-related materials are unequivocally a dynamic component of the global carbon cycle and carbon budget. The characterization of the global carbon cycle and carbon budget is improved by modeling the sponge effect along the cement cycle and its resulting net $CO_2$ emissions balance, especially the $CO_2$ uptake capacity of cement-related materials[5,43]. Notwithstanding that $CO_2$ emissions and $CO_2$ uptake are both subject to substantial uncertainties across scenarios and parameters, it is clear that $CO_2$ uptake will become increasingly significant as cement stocks develop in the future (Fig. 4a). Without a rapid and comprehensive application of mitigation measures in the global cement cycle—even with $CO_2$ uptake accounted for—decarbonizing the cement sector to achieve contemporary climate change goals[38] will remain extremely challenging. Under the nine stock-driven scenarios, the cumulative $CO_2$ uptake (80.4–116.4 Gt) in the global cement cycle from 2015 to 2100 can prevent additional warming of the atmosphere by 0.056–0.081 °C (Fig. 4a), assuming that an increase of 1000 Gt in

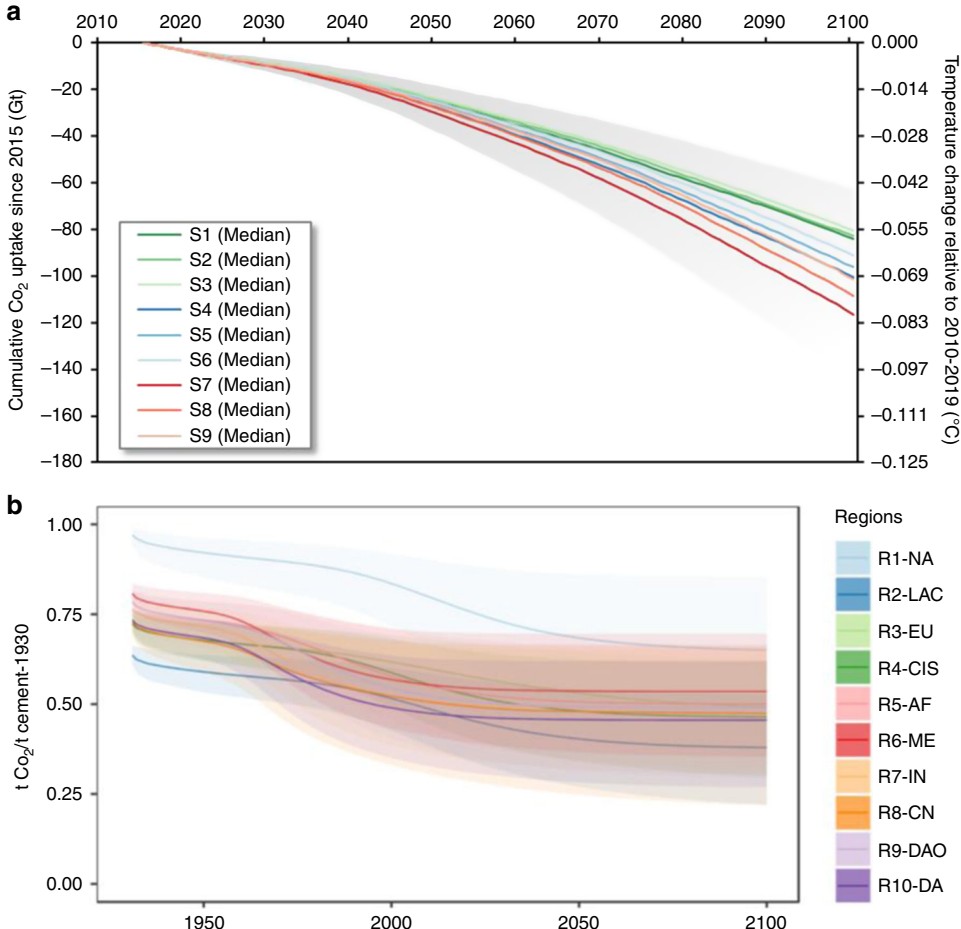

**Fig. 4 Dynamics of cumulative CO₂ uptake in nine stock-driven scenarios and impulse response function of one-tonne cement in ten regions. a** The global cumulative CO₂ uptake from 2015 onward to 2100 in the nine stock-driven scenarios (S1–9) and its consequence on preventable warming relative to the period 2010–2019, based on a warming function (0.693 °C per 1000 Gt CO₂) of cumulative CO₂ emissions derived from ref. [44]. **b** The impulse response function of one-tonne cement produced in 1930 over time (1930–2100) in the ten regions (R1–10). The solid lines are the median value of simulated outcomes, and the shaded areas represent the 95% uncertainty range of simulated outcomes. The number of simulation runs is 1000.

cumulative total anthropogenic CO₂ emissions from 2015 onward would lead to a temperature increase of 0.693 °C[44].

Our analysis reveals that the sponge effect and its resulting net CO₂ emissions balance in the cement cycle resemble the long-term atmospheric decay mechanism of CO₂ emissions exhibited in biomass combustion (i.e., CO₂ emissions from biomass combustion are gradually re-captured by biomass regrowth[45] and thus net emissions tend to zero). The atmospheric decay mechanism of biomass CO₂ emissions is described by an impulse response function, assuming that CO₂ emissions are a pulse and gradually reabsorbed over time[46]. Likewise, a single pulse of CO₂ emissions arising from one tonne of cement produced in 1930 could be represented by an impulse response function (see Fig. 4b). The modeled results are sensitive to regional variations because production technology, use, and fates of cement-related materials are regionally heterogeneous (see details in Supplementary Source Data). The dual interaction of CO₂ emissions and uptake for the cement cycle, which we call the sponge effect, is a nonlinear amortization function over a specific timeframe. This suggests that the timeframe choice for evaluating decarbonization strategies in this material system is of high relevance to life-cycle assessment of cement-related materials. These insights should be consistently included in life cycle assessment studies[47], climate models, and mitigation strategies[38] to facilitate long-term mitigation of CO₂ emissions in the global cement cycle.

## Methods

**Modeling framework**. Modeling procedures and data sources are all delineated in Supplementary Information. The multilayer model used in this study integrates a dynamic material flow analysis model[1,19], the International Energy Agency's global cement technology roadmap[9,24], and a physicochemical model[5] of cement carbonation.

**Dynamic material flow analysis model**. This model consists of two parts: a historical estimation (see details in Supplementary Note 2.1) and a future simulation (see details in Supplementary Note 2.2). The historical cement stocks and flows (1931–2014) are estimated using a top-down stock-flow estimation approach[1]. Future cement flows are simulated using a stock-driven approach[17] and driven by the patterns of in-use stocks (see details in Supplementary Note 2.3 and 2.4) and their lifetimes. Country-specific modeling of the cement cycle requires country-specific assumptions on future stock development, whereas global modeling could not reflect the discrepancies between industrialized and emerging regions. Besides, pairing the country-specific cement cycle with the other two layers requires relevant country-specific understanding. As a compromise, 184 countries are aggregated into ten regions (i.e., North America, Latin America & Caribbean, Europe, Commonwealth of Independent States, Africa, Middle East, India, China, Developed Asia & Oceania, and Developing Asia), each comprising countries with similar socioeconomic and geographic circumstances.

Future pathways for CO₂ emissions and uptake are determined by the dynamics of cement stocks and the material efficiency of new construction, where these patterns enable or constrain the prospects for decarbonization in the global cement cycle. Due to the longevity of in-use cement stocks, demand and demolition are phase displaced[17]: the time of demolition lags behind the original demand by the lifetime of the stock. One of the fundamental assumptions in our scenarios is a moderately growing population, meaning that cement demand and demolition and associated CO₂ emissions and uptake would be significantly affected by population

(see Supplementary Figs. 34–54) and lifetime (see Supplementary Figs. 55–74). Our scenarios encompass a spectrum of material efficiency strategies by varying the level at which in-use cement stocks are assumed to saturate; therefore, greater material efficiency implies reduced levels of stock saturation.

**Material efficiency strategies**. Several material efficiency strategies can be implemented throughout the cement cycle: clinker substitution (M4; e.g., substituting Portland cement clinker for industrial by-products, calcined clay, limestone, etc.); optimizing the cement content of concrete; post-tensioning floor slabs; using more precast building elements; reducing construction waste; avoiding overdesign in construction[41]; reducing cement stock while providing the same level of service (via human settlement design and intensifying use of existing stocks)[48]; lifetime extension[49] (especially important for China due to its current short-lived buildings, see Supplementary Fig. 75). Such measures could fundamentally decouple cement use from service provision, save substantial amounts of $CO_2$ emissions, and thereby lower the probability of infrastructure lock-ins[48]. The implementation of material efficiency strategies is represented by the variation in stock saturation levels across scenarios of roughly 17% (see Supplementary Table 2). Other studies of material efficiency potential have found that greater savings would be possible (see "Discussion"), and thus our scenarios represent a modest level of effort.

**Cement technology roadmap**. The cement technology roadmap outlines five distinct supply-side reduction levers currently available to the cement industry (see details in Supplementary Information Section 3): thermal efficiency (M1), electric efficiency (M2), alternative fuel (M3), clinker substitution (M4), and carbon capture and storage (M5). Thermal efficiency and electric efficiency measures aim at deploying state-of-the-art technologies in new capacities and retrofitting energy-efficient equipment when economically viable. The alternative fuels measure aims to replace fossil fuels by fuels with a higher share of biogenic wastes. The clinker substitution measure seeks to reduce the cement-to-clinker ratio by substituting Portland cement clinker with minerals that have cementitious properties, such as fly ash, ground granulated blast furnace slag, calcined clay, etc. The carbon capture and storage measure aims to capture $CO_2$ as it is emitted, compressing it into a liquid, and storing it in deep underground reservoirs. The International Energy Agency technology roadmap for the global cement industry[4,9] is used to calculate potential reductions in the net $CO_2$ emissions balance of the cement cycle for each of the five mitigation measures.

**Cement carbonation model**. We use a physicochemical model[5] to describe cement carbonation and estimate $CO_2$ uptake during the production, use, and end-of-life stages of the global cement cycle (see details in Supplementary Note 4). In summary, the model takes into account the thicknesses of different cement-related materials, exposure conditions in all life-cycle stages, and atmospheric $CO_2$ concentrations in different regions. To be consistent with the dynamic MFA model, we tailor the physicochemical carbonation model using a survival function[49,50], rather than an average lifetime[5]. Using a survival function captures the survival probability of a group of buildings and infrastructures[23,51], which gives a more reliable measure of the $CO_2$ uptake along the cement cycle.

The total $CO_2$ uptake consists of four sources: cement kiln dust generated from the production stage, construction waste, in-use cement stocks, and demolition waste. Uptake of $CO_2$ by construction cement waste and cement kiln dust is estimated using their generation rates and carbonation fraction. The carbon uptake by concrete and mortar is determined by carbonation rate, CaO content, proportion of CaO that converts to $CaCO_3$ (at complete carbonation), and mole ratio of $CO_2$ to CaO. The carbonation rates are explicitly modeled using Fick's diffusion law. Carbonation rates of in-use concrete and in-use mortar are adjusted by considering the effects of exposed surface area, thickness, compressive strength class, exposure condition, cement additive, atmospheric $CO_2$ concentration, coating and covering, as well as exposure time. Carbonation rates of demolished concrete are modeled, assuming a spherical shape for waste particles. Carbonation rates of demolished mortar are determined by its utilization.

**Calculation of mitigation rate consistent with the 1.5 °C budget**. We calculated $CO_2$ emissions pathways of the cement industry that are consistent with the 1.5 °C budget in the IPCC's special report, following the method employed in refs. [52,53]. We used a budget of 420 Gt (a 66.7% probability of limiting warming to 1.5 °C; see Table 2.2 in ref. [54]) to determine the mitigation rates of $CO_2$ emissions. We assumed that the cement industry is to contribute to the 1.5 °C limit in proportion with other industrial sectors, thereby taking the same mitigation rates (see Supplementary Source Data).

**Limitations and uncertainty**. Although it differentiates the discrepancies among different regions, the global ten-region model can be further improved if country-specific assumptions are available. Beyond this, the main sources of uncertainty are first in the global stock-flow model, and second in the cement carbonation model. The first is mainly accounted for through the range of saturation times and levels in the nine scenarios. The effect of different population forecasts is also explored through sensitivity analysis (see Supplementary Figs. 34–54). For the second set of

uncertainties about the cement carbonation effect, we employed the same Monte Carlo method and parameters used in the global cement carbonation model[5] to estimate uncertainties in $CO_2$ uptake. Critical causes of uncertainties associated with carbonation were identified, and their impacts on simulation results were evaluated by the Monte Carlo method recommended by the 2006 IPCC guidelines for National Greenhouse Gas Inventories[55] (see Supplementary Figs. 76–85). Likewise, we employed the Monte Carlo method to estimate uncertainties in $CO_2$ emissions following the practice recommended by the 2006 IPCC guidelines[55] (see Supplementary Figs. 86–95). The Monte Carlo simulation has been run 1000 times. $CO_2$ emissions from the manufacturing of concrete and mortar and construction of buildings and infrastructures are excluded in the model because they are difficult to allocate to a single material.

**Reporting summary**. Further information on research design is available in the Nature Research Reporting Summary linked to this article.

## Data availability
The data that support the findings of this study are available from the Supplementary Information and Supplementary Source Data. $CO_2$ emission intensities of cement production are available from Getting the Numbers Right (GNR) database (https://www.wbcsdcement.org/GNR-2016/index.html). Data and parameter uncertainties for the physicochemical carbonation model are available from the global cement carbonation model[5]. Source data are provided with this paper.

## Code availability
Data analyses are conducted in R (version 3.5.2) and Excel (version 2016). The R codes used to generate the results on cement carbonation and $CO_2$ emissions presented in this study are available from the following link: https://doi.org/10.5281/zenodo.3384828. The Sankey diagram in Fig. 1 was generated by Circular Sankey developed by Industrial Ecology Freiburg (http://www.visualisation.industrialecology.uni-freiburg.de/). Source data are provided with this paper.

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

## Acknowledgements

This work was financially supported by National Natural Science Foundation of China (71991484), Independent Research Fund Denmark (CityWeight and ReCAP), the Lighthouse ODEx funding of University of Southern Denmark (Building Passport), and the Engineering and Physical Sciences Research Council of the UK (EP/S006079/1 and EP/S019111/1). We are grateful to Lynn Price for comments on an earlier draft of the paper. G.L. thanks his baby boy Huanhuan who kept him up at night to complete the final edits.

## Author contributions

G.L. designed the research. G.L. and Z.C. conceived the paper, developed the model, and collected the data. Z.C. ran the simulation and drew the figures. H.D., T.R.M., and Q.S.G. contributed to conceptual model development. R.J.M., R.S., R.C.L., N.Z., and J.M.C. enhanced cement chemistry and material efficiency discussion. All authors contributed to discussing the results and writing the paper.

## Competing interests

The authors declare no competing interests.
