## [Peer Review File · Nature Communications]

Reviewers' comments:

Reviewer #1 (Remarks to the Author):

This is a well-prepared manuscript covering a critical topic in the assessment of CO₂ emissions associated with cement production, use, and disposal. The manuscript draws attention to projected carbonation in cement-based materials, which is not typically considered in the discussion of mitigation methods for CO₂ emissions from the cement industry. As such, this work will be mostly of relevance to those interested in CO₂ mitigation in building materials or in the industrial sector (for which, cement production is a notable contributor to CO₂ emissions). The work expands upon findings of Xi et al. 2016 <https://www.nature.com/articles/ngeo2840>. It is constructed from several sources, but predominantly builds from a published in-use stock model for cement globally (by some of the authors on this manuscript) and from a published model (that by Xi et al. 2016) for the carbonation of cement-based materials globally. The authors develop projection scenarios implementing these models and present corresponding findings.

Overall, I would not go so far as to say the work is novel or outstanding, but it is a valuable contribution to the literature. There are several of improvements that should be made to the manuscript and its supporting information.

There is a need for some editorial improvements. These range from minor grammatical issues (e.g., "include production, use..." should be "including production, use..." in the abstract) to several of the figures in the supporting information lacking axis labels or figure keys or both, making them difficult to interpret. These are examples of issues that are present in the work and the reviewer trusts that the authors will correct such errors throughout the manuscript and its supporting information.

The clarity of presentation of assumptions made and results found could be improved. For example, the writing style used in lines 110 to 121 is quite hard to follow. I found Figure 3 to be unclear; a reader has to look very carefully for differences in colors used and the labeling leads to switching back and forth from the caption to the figure as opposed to being able to clearly read the figure. Quantitative reasons for selecting the saturation levels and rates used in the projection models should be improved. Expanded discussion of application of models to develop figures presented in the supporting information should be provided. Uncertainties in the model are only discussed for the carbonation aspect; uncertainties for other aspects should be considered and articulated (e.g., CO₂ emission variations from different fossil fuels – this is simply one example). From the discussion on lines 70-71 and the supporting information, it is unclear the extent to which emissions from raw material acquisition and transport of materials is considered prior to the completion of the cement production. Reference to cement-equivalents (e.g., Figure 1 caption) is made without a definition or explanation as to how these were determined. It is unclear as to how uncertainties in the mitigation strategies were considered beyond the sensitivity analysis of varying saturation rate and level. For the Methods, it would help immensely to have more specific references to individual sections in the supporting information where the reader can find the quantitative details; as written, the methods are very vague and the details are buried in the supporting information. The abstract could be punchier, perhaps by adding some more quantitative takeaways.

Other more minor issues:

- Is there a typo in line 124? There is reference to the late 2100s, but the study seems to end at 2100. In the same paragraph, it is unclear what "below the 2010 level ... relative to the 2010 level" means.
- Which CCS technology is modeled? Which mineral admixtures are modeled?
- Lines 333 – 334, it would help the reader to have a citation or an explanation as to why the model selected is more reliable.

I did not find any issues with the statistics presented. Where possible, I suggest adding in all data used (recognizing some data may be under copyright and not easily shared). As it currently stands, the reader is sent to many disparate sources to piece together data inputs used for models. Additionally, the authors should provide access to the codes written to perform the analyses presented. Availability of data and codes are required by the Nature journals and making such information available is proper scientific conduct.

Reviewer #2 (Remarks to the Author):

This paper describes linking a set of models for the projection of demand for cement and supply technologies with a module that calculated the uptake of CO₂ by demand during its lifetime in buildings and infrastructure. In doing so, it integrates the methodology that Xi et al introduced in 2016 within the broader discussion on the mitigation of emissions from the cement sector. This is an extremely important step forward in assessing the role of the cement sector in greenhouse gas mitigation scenarios.

In general I certainly recommend that this paper should be published in Nature Communications, but I have a few suggestions to improve its impact:

1. The current debate on climate policy is on the remaining budget for reaching 1.5 degree global warming. This budget is very small, around 600 GtCO₂ for a 50% chance (see IPCC SR1.5, Chapter 2, Table 2.2). Putting this analysis of the cement sector in the context of such small budget would greatly enhance the policy relevance of the paper. A difference of 100-140 GtCO₂ in the uptake of CO₂ by cement in buildings would make a large difference in this discussion.
2. The use of wedges terminology seems rather outdated. First of all, the paper that introduced the wedges was published a long time ago, but it also became clear shortly afterwards that distinguishing between the different wedges isn't that straightforward. I would recommend to replace the wedges terminology with a simple description of technology/mitigation options.
3. The authors briefly mention a comparison of their demand projections with the IEA for 2050. However, since they present a completely new cement demand model based on stocks, it would be good to place their projections of cement demand in the broader context of the literature.
4. Figure 4b needs more thorough explanation. If this is a model for the one time impulse emissions from the production of 1 tonne of cement in 2030, why are the patterns between the regions so extremely different? In some regions, the "tCO₂/t-cement 2030" drops quickly around 1960, whereas in others such drop occur only ~70 years after the cement was produced. What causes this difference in timing between the regions?

Reviewer #3 (Remarks to the Author):

This is a thorough study by a team of well-acknowledged scholars in this field. Clearly, a lot of effort has gone into this work. However, several issues are evident which the authors need to carefully revisit. While it is an interesting and detailed example of a scenario-based material stock & flow study of cement, I'm afraid that the authors should be more careful in interpreting the significance and robustness of their results, and that therefore the manuscript fails to achieve Nature Communications' criteria for publication. Namely, that the paper provides strong evidence for its conclusions, that the results are truly novel, and that it represents an advance in understanding likely to influence thinking in the field. I detail my misgivings below.

1. I am concerned that there's a mismatch between the presentation of this study's novelty and significance in the manuscript to its actual novelty and significance. First, the selection of references that this study acknowledges, as well as the style and choice of words in which they are referred to, give the impression that the authors operate nearly alone in this space. This is

probably unintended and a poor choice of words perhaps to save on word count, but is nevertheless inappropriate especially for such a high profile journal like Nature Communications. There's nominal mention of current & past research into cement cycles and environmental consequences – even of studies by the authors themselves! – and no descriptions of their findings. Likewise there's minimal mention of previous studies of carbon absorption by cement and their findings, and no discussion of comparisons of past studies' results to this study. This can be remedied by a careful discussion of the state of the art.

2. However, to a greater extent the issue of novelty and significance is reflected in the fact that this study is in many ways simply an extension of the authors' previous studies: greater scope, multiple scenarios, and extended to 2100. The authors should clearly explain how this extra effort adds new understanding and new insights. However the authors don't describe the aims of this work, nor do they explain how their new results would contribute to our understanding of the subject, so it's hard to understand what their intentions are. My understanding is that their take-home messages are threefold: (1) that cement plays a big role in CO₂ emissions and will continue to do so, (2) material efficiency measures can help to an extent but CCS will be necessary to truly mitigate emissions from this sector, and (3) previous studies have neglected the future "passive sequestration / sponge effect". Findings 1 and 2 are quite well known from past research and in this regard the novel contribution of this study is in new future estimates (but see the next comments about their robustness). Therefore I suppose it's no surprise that the authors choose to focus in the title and abstract on finding 3 and indeed this comparison of future "passive sequestration" to CCS/"active sequestration" is intriguing. Yet in the paper itself, they only refer to what they term the "sponge effect" late in the text – only in page 7 – and spend very little time discussing it before reverting to other topics. Perhaps because it turned out not to be so significant? There's therefore a mismatch between what the title and abstract suggest and what the actual manuscript presents.

3. The study hinges on the plausibility of the scenario formulations and how these are expressed through the models & assumptions. The authors need to do more to be able to claim that their future scenario results are "stark and robust" (line 203). They set an ambitious time horizon to 2100, 80 years during which it is plausible that not only step-changes but fundamental regime shifts and structural breaks in technologies, socio-economics, and so forth would occur. However, despite formulating 9 scenarios, the scenario space is actually quite narrow and restricted by explicit and implicit scenario and modeling assumptions which are quite conservative and inflexible. In consequence this modeling exercise ends up being less ambitious and comprehensive than the authors seem to describe. The authors should either revisit their assumptions and expand on them, or rephrase the study in a more careful fashion to ensure that they don't accidentally overstate the interpretations of the results.

4. For example, perhaps the most influential parameter in the model is population (as nearly all other parameters are coefficients) but it has only one future pathway based on a UN scenario that is taken for granted without any analysis or comparison of alternatives and its effects on the results. This is in itself a legitimate what-if assumption but needs to be described outright, together with the limitations it induces in the results and their interpretations: some text like "we model a scenario set based on a single UN population forecast, to explore the consequences of such a population growth scenario on flows and stocks of cement & the related CO₂ emissions. Our results should therefore be interpreted within this context". Similar text should be included for all other assumptions, to avoid misrepresentation and accidental overconfidence of the future results.

5. According to the SI, the per-cap s-curves' inflection points are modeled to occur in the future – meaning that, by definition, peaks of inflows are forced to happen in the future (rather than already have happened in the past). The authors don't present any plausibility checks for this assumption even for regions in which the historical pattern already visually suggests that the inflection point has already occurred (eg. North America, Europe, and most pronounced in Developed Asia & Australia). These peaks seem to be rather dramatic, especially in the "fast"

scenarios, and permeate throughout the model (e.g. they also determine the scale of demand for the next replacement cycles), and I suspect they're the cause of the peculiar global fluctuations described in lines 110-112 and figure 2. It's critical to check if this assumption is plausible and ascertain the scale of its influence over the results. If the authors choose to keep this assumption, then some text like "we assume that no region has reached peak cement inflows" etc. should be included.

6. If I understand correctly, in the 9 main scenarios the authors assume that building & infrastructure lifetimes will not change until 2100. Yet empirically it has been shown to have changed throughout the 20th century. Besides being a strong assumption, it has major implications on the results: on the one hand it forces regions with short lifetimes like china to require rapid cycles of cement inflows for stock maintenance, and at the same time gives them an "unfair" advantage by being able to adapt best practices of material efficiency strategies more often due to having less lock-in effects from older stocks. In short, the lifetime assumption influences the results in multiple significant ways. There's limited and insufficient acknowledgement of this (only mentioned in pages 80-81 of the SI and even then without any real discussion of the consequences, and only for China), but this assumption should be revisited and thoroughly described.

7. I'm also concerned about the choice of 2100 as the time horizon: I suspect that it's not because of the insights into the potential future (after all, forecasting 80 years into the future is virtually meaningless – consider scenarios and forecasts from the 1940s of today) but rather that the choice of 2100 was made because the scenario setups and modeling assumptions cause most of the "interesting" things to happen after 2050, like the saturations, peaks and subsequent cycles of demand, and so forth. Hence the 2100 results tell us less about what to prepare for in the future, and more about how the model operates. The authors neglect to acknowledge this modeling artefact, and instead present their 2100 results as a future of some likelihood and seeming confidence, which risks being misunderstood by less diligent readers.

8. Related to these issues, the authors present numbers with decimal-digit precision (e.g. 58.4%, 101.6 Gt...), where prudence would allow perhaps for rounding up to whole integers or even tens or hundreds (e.g. ~60%, ~100 Gt). This is relevant to all numbers in the text.
Good luck!

Response to reviewers for manuscript NCOMMS-19-14627-T

“The sponge effect and carbon emission mitigation potentials of the global cement cycle”

Reviewer #1:

Comment #1

This is a well-prepared manuscript covering a critical topic in the assessment of CO₂ emissions associated with cement production, use, and disposal. The manuscript draws attention to projected carbonation in cement-based materials, which is not typically considered in the discussion of mitigation methods for CO₂ emissions from the cement industry. As such, this work will be mostly of relevance to those interested in CO₂ mitigation in building materials or in the industrial sector (for which, cement production is a notable contributor to CO₂ emissions). The work expands upon findings of Xi et al. 2016 <https://www.nature.com/articles/ngeo2840>. It is constructed from several sources, but predominantly builds from a published in-use stock model for cement globally (by some of the authors on this manuscript) and from a published model (that by Xi et al. 2016) for the carbonation of cement-based materials globally. The authors develop projection scenarios implementing these models and present corresponding findings.

Thanks for your comments and comprehensive assessment.

Comment #2

Overall, I would not go so far as to say the work is novel or outstanding, but it is a valuable contribution to the literature. There are several of improvements that should be made to the manuscript and its supporting information.

We have considered all your comments and improved our manuscript.

Comment #3

There is a need for some editorial improvements. These range from minor grammatical issues (e.g., “include production, use...” should be “including production, use...” in the abstract) to several of the figures in the supporting information lacking axis labels or figure keys or both, making them difficult to interpret. These are examples of issues that are present in the work and the reviewer trusts that the authors will correct such errors throughout the manuscript and its supporting information.

We have carefully proofread the main manuscript and the supplementary information. Please find the edits we made in the track-changes version.

We have updated figures in the Supplementary Information, including adding axis labels to Figures S2-S11, S12-S22, S23-S33, S34-S40, as well as a figure key to Figures S44-S83.

Comment #4

The clarity of presentation of assumptions made and results found could be improved. For example, the writing style used in lines 110 to 121 is quite hard to follow.

We have thoroughly edited the manuscript to guarantee that the results are clearly delivered.

We have rephrased this paragraph to clarify the definition of “no-action CO₂ emissions” and “no-action CO₂ emissions balance”. A few words are added in the caption of Figure 1 to explain the three lines.

The presentation of results (Figure 2) has been improved.

See lines 200-212 (track-changes version):

“Fig. 2a shows CO₂ emissions under the ‘no-action’ scenario and the effects of the mitigation measures. In 2050, the ‘no-action’ CO₂ emissions under low, medium, and high saturation levels reach 3.7-4.6 Gt yr⁻¹, 4.0-5.0 Gt yr⁻¹, and 4.2-5.5 Gt yr⁻¹, respectively. In parallel, the CO₂ uptake (effects of U-M4 subtracted, the same hereinafter) rises to 1.1-1.3 Gt yr⁻¹ (low saturation levels), 1.1-1.4 Gt yr⁻¹ (medium saturation levels), and 1.2-1.5 Gt yr⁻¹ (high saturation levels) by 2050. The ‘no-action’ CO₂ emissions balance (when CO₂ uptake is considered) in 2050 increases by a factor of 1.3-1.7 (low saturation levels), 1.5-1.9 (medium saturation levels), and 1.6-2.1 (high saturation levels), respectively, compared to the 2010 level of 1.9 Gt yr⁻¹. By 2100, the balance reaches 3.0-3.2 Gt yr⁻¹ (low saturation levels), 3.3-3.6 Gt yr⁻¹ (medium saturation levels), and 3.7-4.0 Gt yr⁻¹ (high saturation levels), respectively.”

I found Figure 3 to be unclear; a reader has to look very carefully for differences in colors used and the labeling leads to switching back and forth from the caption to the figure as opposed to being able to clearly read the figure.

We have replotted Figure 3. In order to make the share of different mitigation measures more distinct, we use column charts to present their relative contribution.

Quantitative reasons for selecting the saturation levels and rates used in the projection models should be improved.

We have restructured the texts and added a few lines to clarify our assumptions on saturation levels and rates.

See lines 113-134 (track-changes version):

“Using a top-down stock-flow approach driven by data on cement production, trade, sectoral use, and lifetime¹, we estimate the historical and contemporary cement stocks. We observe that the per capita cement stocks in all ten regions have increased since 1930 (see Supplementary Fig. S2-11). Global average cement stocks per capita reached 10.2 t in 2014, with industrialized and transitioning regions ranging from 12.7 to 23.7 t, developing regions ranging from 2.7 to 7.5 t, and several mature economies approaching 35 t. However, regional cement stocks are not equally distributed across sectors. Post-industrial regions (especially the Commonwealth of Independent States; CIS) typically have higher levels of per capita cement stocks in the civil engineering sector. In contrast, China has a lower level of per capita cement stocks in the civil engineering sector but a considerably higher level in buildings. We speculated that these variations could be explained by multiple factors, such as development stage, patterns of urban expansion, architectural specification, as well as availability and choice of construction materials¹. Earlier studies have shown a saturation phenomenon for per capita in-use stock development of bulk materials, such as iron^{25,26} and copper²⁷ in industrialized countries, but not for aluminum, due to its relatively short history of use²⁸. Likewise, the development patterns of per capita cement stocks generally comply with an S-shaped curve, and saturation is evident in several highly-developed countries¹. The saturation of per capita cement stocks implies that the growth rate of buildings and infrastructures (where cement stocks reside) will decrease marginally and eventually reach a plateau, as services provided by cement stocks become saturated^{16,29–32}. Concurrently, demand for cement will slow down and ultimately stabilize. We therefore infer that the dynamics of in-use cement stocks, to a large degree, determine the demolition rate and reconstruction rate for cement-related materials, according to the mass-balance principle^{20,33}. In addition, we use the level of in-use cement stocks as an explicit physical representation of service provision to society²⁰.”

Expanded discussion of application of models to develop figures presented in the supporting information should be provided.

We have added some lines to detail how the models operate.

See lines 312-318 in Supplementary Information (track-changes version):

“The stock-driven approach was employed to simulate inflows and outflows from 2015 to 2100. Annual cement inflows were driven by changes in cement stocks and annual cement outflows, as shown in Eq.7. The changes in cement stocks were determined by population and per capita cement stock. Annual cement outflows were determined by historical cement inflows and lifetime. The stock-driven simulations were conducted by sectors and the sectoral outputs were subsequently used to calculate the CO₂ emissions and CO₂ uptake along the entire cement cycle. Intermediate outputs from the simulations are shown in the following figures.”

Uncertainties in the model are only discussed for the carbonation aspect; uncertainties for other aspects should be considered and articulated (e.g., CO₂ emission variations from different fossil fuels – this is simply one example).

We have considered uncertainties in the CO₂ emissions, including variations in process emission factor, clinker-to-cement ratio, thermal efficiency, thermal emission factor, electric efficiency, and electric emission factor (See lines 478-493 in Supplementary Information). Results of uncertainties in the CO₂ emissions are presented in Section 5.1 of Supplementary Information.

From the discussion on lines 70-71 and the supporting information, it is unclear the extent to which emissions from raw material acquisition and transport of materials is considered prior to the completion of the cement production.

We have added some lines to clarify the scope of CO₂ emissions accounting.

See lines 465-471 in Supplementary Information (track-changes version):

“GNR database consists of data collected from individual companies. According to CSI’s reporting standard¹⁷, direct CO₂ emissions occurring from sources that are owned or controlled by the company and indirect CO₂ emissions from the generation of purchased electricity consumed in the company’s owned or controlled equipment are included in the CO₂ emissions accounting. CO₂ emissions from off-site transports of mineral inputs and products are not included. These off-site CO₂ emissions are typically small and difficult to quantify consistently, because these transports are often carried out by third parties.”

Reference to cement-equivalents (e.g., Figure 1 caption) is made without a definition or explanation as to how these were determined.

We have added some explanation to clarify how cement-equivalent was defined.

See the caption of Figure 1.

It is unclear as to how uncertainties in the mitigation strategies were considered beyond the sensitivity analysis of varying saturation rate and level.

Beyond the sensitivity analysis of varying saturation rate and level, we have additionally conducted sensitivity analyses on the impact of lifetime on CO₂ emissions and CO₂ uptake.

See Section 5.3 of Supplementary Information.

For the Methods, it would help immensely to have more specific references to individual sections in the supporting information where the reader can find the quantitative details; as written, the methods are very vague and the details are buried in the supporting information.

We have specified which section is referred to.

See edits throughout the manuscript.

The abstract could be punchier, perhaps by adding some more quantitative takeaways.

We have edited the abstract to make it punchier with more quantitative take-away messages.

See lines 24-38 (track-changes version):

“Cement plays a dual role in the global carbon cycle like a sponge: its massive production contributes significantly to present-day global anthropogenic CO₂ emissions, yet its hydrated products gradually reabsorb substantial amounts of atmospheric CO₂ (carbonation) in the future. The role of this sponge effect along the cement cycle (including production, use, and demolition) in carbon emissions mitigation, however, remains hitherto unexplored. Here, we quantify the effects of demand- and supply-side mitigation measures considering this material-energy-emissions-uptake nexus, finding that climate goals would be imperiled as global in-use stocks of cement develop. Future reabsorption of CO₂ will be significant (~30% of cumulative CO₂ emissions from 2015 to 2100), but climate goal compliant net CO₂ emissions reduction along the global cement cycle will require both radical technology advancements (e.g., carbon capture and storage) and widespread deployment of material efficiency measures, which go beyond those envisaged in current technology roadmaps.”

Comment #5

Other more minor issues:

- Is there a typo in line 124? There is reference to the late 2100s, but the study seems to end at 2100. In the same paragraph, it is unclear what “below the 2010 level ... relative to the 2010 level” means.

We have double checked the texts and corrected the typos throughout the manuscript.

- Which CCS technology is modeled? Which mineral admixtures are modeled?

We have added a few lines to specify which CCS technologies are modeled.

See lines 589-595 in Supplementary Information (track-changes version):

“Chemical absorption with amine-based solvents is a promising post-combustion technology, because its operational experiences are available from several industries (e.g., chemical and gas industry). In the long run, membrane technologies seem to be a candidate, while physical absorption or mineral carbonation seems to be less feasible due to lack of sustained mass streams of sorbents^{19,20}. Oxyfuel technology aims to generate a comparatively pure CO₂ stream by using oxygen instead of air in the cement kiln firing and thus the purified CO₂ streams could be transported or stored with less effort^{19,20}.”

Detailed description of which minerals are used to substitute clinker is added in Supplementary Information.

See lines 571-574 in Supplementary Information (track-changes version):

“Portland cement clinker is finely inter-ground with gypsum to control its setting properties and is also sometimes blended with other (cementitious) materials to further modify performance, including blast furnace slag, fly ash, limestone, and natural volcanic materials.”

- Lines 333 – 334, it would help the reader to have a citation or an explanation as to why the model selected is more reliable.

We added a few words to explain why a survival function is used.

See lines 462-463 (track-changes version):

“Using a survival function captures the survival probability of a group of buildings and infrastructures^{22,49}, which gives a more reliable measure of the CO₂ uptake along the cement cycle.”

Comment #6

I did not find any issues with the statistics presented. Where possible, I suggest adding in all data used (recognizing some data may be under copyright and not easily shared). As it currently stands, the reader is sent

to many disparate sources to piece together data inputs used for models. Additionally, the authors should provide access to the codes written to perform the analyses presented. Availability of data and codes are required by the Nature journals and making such information available is proper scientific conduct.

We have added two sections: Code Availability and Data Availability. We have added the link (zenodo) for computer codes used for generating the results on cement carbonation and CO₂ emissions.

Reviewer #2 (Remarks to the Author):

Comment #1

This paper describes linking a set of models for the projection of demand for cement and supply technologies with a module that calculated the uptake of CO₂ by demand during its lifetime in buildings and infrastructure. In doing so, it integrates the methodology that Xi et al introduced in 2016 within the broader discussion on the mitigation of emissions from the cement sector. This is an extremely important step forward in assessing the role of the cement sector in greenhouse gas mitigation scenarios.

In general I certainly recommend that this paper should be published in Nature Communications, but I have a few suggestions to improve its impact:

Thanks for your positive feedbacks. We have carefully addressed all the comments you raised.

Comment #2

1. The current debate on climate policy is on the remaining budget for reaching 1.5 degree global warming. This budget is very small, around 600 GtCO₂ for a 50% chance (see IPCC SR1.5, Chapter 2, Table 2.2). Putting this analysis of the cement sector in the context of such small budget would greatly enhance the policy relevance of the paper. A difference of 100-140 GtCO₂ in the uptake of CO₂ by cement in buildings would make a large difference in this discussion.

We have incorporated the 1.5 degrees budget in our study. We present the results of a budget of 420 Gt in Figure 2. The results of a budget of 580 Gt are available in Supplementary Source Data.

See lines 477-483 (track-changes version):

“Calculation of CO₂ emissions pathways consistent with the 1.5 °C budget. We calculated CO₂ emissions pathways of the cement industry that are consistent with the 1.5 °C budget in the IPCC’s special report, following the method employed in ref. ^{50,51}. We used a budget of 420 Gt (a 66.7%

probability of limiting warming to 1.5 °C; see table 2.2 in ref. ⁵²) to determine the mitigation rates of CO₂ emissions. We assumed that the cement industry is to contribute to the ‘1.5 °C limit’ in proportion with other industrial sectors, thereby taking the same mitigation rates (see Supplementary Source Data).”

Comment #3

2. The use of wedges terminology seems rather outdated. First of all, the paper that introduced the wedges was published a long time ago, but it also became clear shortly afterwards that distinguishing between the different wedges isn’t that straightforward. I would recommend to replace the wedges terminology with a simple description of technology/mitigation options.

We have replaced wedge terminology.

See edits throughout the manuscript and supplementary information.

Comment #4

3. The authors briefly mention a comparison of their demand projections with the IEA for 2050. However, since they present a completely new cement demand model based on stocks, it would be good to place their projections of cement demand in the broader context of the literature.

We have incorporated the IEA new report “Material efficiency in clean energy transitions”.

See lines 305-311 (track-changes version):

“The significance of material efficiency strategies is also examined in a special report of International Energy Agency technology, in which a bottom-up analysis of the building sector shows that material efficiency improvements in the buildings sector can reduce approximately 26% of its annual cement demand in 2060 (see figure 25 in ref. ⁴⁰). Accounting for both saturation levels and saturation time, in 2060, annual global cement demand sees a 34% decline in the Low-Slow scenario (5.5 Gt yr⁻¹) relative to the High-Fast scenario (8.3 Gt yr⁻¹).”

Comment #5

4. Figure 4b needs more thorough explanation. If this is a model for the one time impulse emissions from the production of 1 tonne of cement in 2030, why are the patterns between the regions so extremely different? In some regions, the “tCO₂/t-cement 2030” drops quickly around 1960, whereas in others such drop occur only ~70 years after the cement was produced. What causes this difference in timing between the regions?

We have added the source data supporting Figure 4b.

The impulse response function of one-tonne cement produced in 1930 over time largely coincides with the lifetime distribution. We plotted a graph on **CO₂ uptake** of one-tonne cement produced in 1930 over time (see the graph attached below; data supporting this graph are documented in Supplementary Source Data, Table Fig.4b) to help understand why patterns of the impulse response function are different. During the use stage, carbonation occurs in the first few years. The demolition rate starts to peak, which leads to increases in CO₂ uptake.

Reviewer #3 (Remarks to the Author):

Comment #1

This is a thorough study by a team of well-acknowledged scholars in this field. Clearly, a lot of effort has gone into this work. However, several issues are evident which the authors need to carefully revisit. While it is an interesting and detailed example of a scenario-based material stock & flow study of cement, I'm afraid that the authors should be more careful in interpreting the significance and robustness of their results, and that therefore the manuscript fails to achieve Nature Communications' criteria for publication. Namely, that the paper provides strong evidence for its conclusions, that the results are truly novel, and that it represents an advance in understanding likely to influence thinking in the field. I detail my misgivings below.

Thanks a lot for your time and thorough assessment and incisive comments. We have tried to integrate your suggestions and revisited the language which was unclear and could lead to misunderstandings. We hope our revision would address your misgivings.

Comment #2

1. I am concerned that there's a mismatch between the presentation of this study's novelty and significance in the manuscript to its actual novelty and significance. First, the selection of references that this study acknowledges, as well as the style and choice of words in which they are referred to, give the impression that the authors operate nearly alone in this space. This is probably unintended and a poor choice of words perhaps to save on word count, but is nevertheless inappropriate especially for such a high profile journal like Nature Communications. There's nominal mention of current & past research into cement cycles and environmental consequences – even of studies by the authors themselves! – and no descriptions of their findings. Likewise there's minimal mention of previous studies of carbon absorption by cement and their findings, and no discussion of comparisons of past studies' results to this study. This can be remedied by a careful discussion of the state of the art.

We have revised the literature view to acknowledge the contribution of previous studies and to clarify the contribution of this study.

See lines 56-76 (track-changes version):

“Although the carbonation effect is well known as a deterioration mechanism of concrete, it has relatively recently been recognized as a potentially significant CO₂ sink⁶. The scale of historical CO₂ absorption has been estimated regionally^{6,8} and globally⁵, concluding that nearly half of process emissions in cement production from 1930 to 2013 have likely since been sequestered⁵. Understanding the mitigation potential of the sponge effect requires looking to the future, but future scenarios are often based on cement demand linked to market growth^{9,10} or economic indicators^{11,12}. However, a proper understanding of the sponge effect requires not just forecasting cement demand but a physically-consistent accounting of the cement stocks in the built environment, the cement demand for replacement and expansion of stocks, and the end-of-life demolition waste.

The global convergence of buildings and infrastructure services in all nations, to the level of industrialized countries, is expected to drive sustained increases in global cement demand to build up the desired in-use stocks^{1,13–15}. Simultaneous expansion, demolition, and replacement of cement stocks in the

built environment will generate significant amounts of demolition waste once buildings and infrastructures reach their end-of-life^{16,17}, as well as construction waste during their construction, both of which have different CO₂ absorption characteristics from cement in active use and account for a large part of the lifetime CO₂ absorption⁶. The use patterns of cement stocks and their longevity (lifetime from decades to centuries) create long-term path dependences for both cement demand and demolition waste generation^{18–21}. The explicit characterization of cement flows and stocks enables an explicit understanding of the components of the sponge effect and the resulting net CO₂ emissions balance along the future cement cycle, which has been missing in previous work.”

Comment #3

2. However, to a greater extent the issue of novelty and significance is reflected in the fact that this study is in many ways simply an extension of the authors’ previous studies: greater scope, multiple scenarios, and extended to 2100. The authors should clearly explain how this extra effort adds new understanding and new insights. However the authors don’t describe the aims of this work, nor do they explain how their new results would contribute to our understanding of the subject, so it’s hard to understand what their intentions are. My understanding is that their take-home messages are threefold: (1) that cement plays a big role in CO₂ emissions and will continue to do so, (2) material efficiency measures can help to an extent but CCS will be necessary to truly mitigate emissions from this sector, and (3) previous studies have neglected the future “passive sequestration / sponge effect”. Findings 1 and 2 are quite well known from past research and in this regard the novel contribution of this study is in new future estimates (but see the next comments about their robustness). Therefore I suppose it’s no surprise that the authors choose to focus in the title and abstract on finding 3 and indeed this comparison of future “passive sequestration” to CCS/“active sequestration” is intriguing. Yet in the paper itself, they only refer to what they term the “sponge effect” late in the text – only in page 7 – and spend very little time discussing it before reverting to other topics. Perhaps because it turned out not to be so significant? There’s therefore a mismatch between what the title and abstract suggest and what the actual manuscript presents.

The “sponge effect” metaphor is two-fold: cement production releases CO₂ emissions like a pulse and cement hydration products gradually reabsorb atmospheric CO₂ afterwards. The characterization of cement flows and stocks enables an explicit understanding of the sponge effect and its resulting net CO₂ emissions balance along the cement cycle.

We have made thorough edits to clarify the role of the sponge effect along the cement cycle in future CO₂ emissions mitigation. We made it clear in the very beginning of the introduction.

See lines 48-52 (track-changes version):

“We refer to this dual role in emitting and soaking up CO₂ along the entire cement cycle (from production, through use, and to end-of-life) as the “sponge effect”, and it must be considered in examining long-term decarbonization pathways and identifying carbon management strategies for this material system.”

Comment #4

3. The study hinges on the plausibility of the scenario formulations and how these are expressed through the models & assumptions. The authors need to do more to be able to claim that their future scenario results are “stark and robust” (line 203). They set an ambitious time horizon to 2100, 80 years during which it is plausible that not only step-changes but fundamental regime shifts and structural breaks in technologies, socio-economics, and so forth would occur. However, despite formulating 9 scenarios, the scenario space is actually quite narrow and restricted by explicit and implicit scenario and modeling assumptions which are quite conservative and inflexible. In consequence this modeling exercise ends up being less ambitious and comprehensive than the authors seem to describe. The authors should either revisit their assumptions and expand on them, or rephrase the study in a more careful fashion to ensure that they don’t accidentally overstate the interpretations of the results.

We have rephrased the assumptions of our scenarios and added a few lines to explain the rationale for formulating the nine scenarios.

See lines 138-159 (tack-changes version):

“In light of the observed historical patterns of cement stocks and the essential role of in-use stock dynamics to the cement cycle, we simulate the future cement cycle in ten regions using a stock-driven approach¹⁶ based on the historical patterns of in-use cement stocks identified in our previous work¹ and a growing population obtained from the medium scenario of United Nations World Population Prospects³⁴. We construct nine stock-driven scenarios to explore the evolution of cement-related materials until 2100 due to the longevity of buildings and infrastructures. Our scenarios are based on the mass balance principle, and assume that in-use cement stocks will eventually saturate in each of the 10 regions analyzed in different socioeconomic contexts (see Supplementary Fig. S1). In all of the nine scenarios, we parameterize two boundary conditions, saturation level and saturation time, to reflect the varying patterns of cement stocks and varying levels of future ‘demand-side’ material efficiency in different regions. The saturation level of in-use cement stocks is regarded as a tangible indicator for various

human needs in mature societies, including shelter, transport networks, factories, offices, as well as commercial, educational, healthcare, and governmental facilities. It is the level of *service* provided by in-use cement stocks that is expected to saturate, not just the quantity of *material* involved; the two are linked by the material intensity of the in-use cement stocks. By considering a range of saturation levels, we cover both a range of service levels provided by the in-use cement stocks and a range of material efficiencies in their delivery. The saturation time indicates when the in-use cement stocks reach 98% of the saturation level. Given the regional heterogeneity of socioeconomic and geographic circumstances, we set varying saturation levels and times for different regions to fit the historical development of their in-use cement stocks (see Supplementary Table S3).”

Comment #5

4. For example, perhaps the most influential parameter in the model is population (as nearly all other parameters are coefficients) but it has only one future pathway based on a UN scenario that is taken for granted without any analysis or comparison of alternatives and its effects on the results. This is in itself a legitimate what-if assumption but needs to be described outright, together with the limitations it induces in the results and their interpretations: some text like “we model a scenario set based on a single UN population forecast, to explore the consequences of such a population growth scenario on flows and stocks of cement & the related CO2 emissions. Our results should therefore be interpreted within this context”. Similar text should be included for all other assumptions, to avoid misrepresentation and accidental overconfidence of the future results.

We have articulated the population assumption.

See lines 141-142 (tack-changes version):

“.....a growing population obtained from the medium scenario of United Nations World Population Prospects³⁴.”

Comment #6

5. According to the SI, the per-cap s-curves’ inflection points are modeled to occur in the future – meaning that, by definition, peaks of inflows are forced to happen in the future (rather than already have happened in the past). The authors don’t present any plausibility checks for this assumption even for regions in which the historical pattern already visually suggests that the inflection point has already occurred (eg. North America, Europe, and most pronounced in Developed Asia & Australia). These peaks seem to be rather dramatic, especially in the “fast” scenarios, and permeate throughout the model (e.g. they also determine the scale of demand for the next replacement cycles), and I suspect they’re the cause of the peculiar global fluctuations described in lines 110-112

and figure 2. It's critical to check if this assumption is plausible and ascertain the scale of its influence over the results. If the authors choose to keep this assumption, then some text like "we assume that no region has reached peak cement inflows" etc. should be included.

We highly agree with your suggestion. We have made our assumptions clearer.

See lines 145-147 (track-changes version):

"Our scenarios are based on the mass balance principle, and assume that in-use cement stocks will eventually saturate in each of the 10 regions analyzed in different socioeconomic contexts (see Supplementary Fig. S1)."

Comment #7

6. If I understand correctly, in the 9 main scenarios the authors assume that building & infrastructure lifetimes will not change until 2100. Yet empirically it has been shown to have changed throughout the 20th century. Besides being a strong assumption, it has major implications on the results: on the one hand it forces regions with short lifetimes like china to require rapid cycles of cement inflows for stock maintenance, and at the same time gives them an "unfair" advantage by being able to adapt best practices of material efficiency strategies more often due to having less lock-in effects from older stocks. In short, the lifetime assumption influences the results in multiple significant ways. There's limited and insufficient acknowledgement of this (only mentioned in pages 80-81 of the SI and even then without any real discussion of the consequences, and only for China), but this assumption should be revisited and thoroughly described.

We have conducted comprehensive sensitivity analyses on three representative regions: North America, Europe, and China.

We have presented intermediate results in Section 5.3 of Supplementary Information to demonstrate how lifetime change would affect the results.

Comment #8

7. I'm also concerned about the choice of 2100 as the time horizon: I suspect that it's not because of the insights into the potential future (after all, forecasting 80 years into the future is virtually meaningless – consider scenarios and forecasts from the 1940s of today) but rather that the choice of 2100 was made because the scenario setups and modeling assumptions cause most of the "interesting" things to happen after 2050, like the saturations, peaks and subsequent cycles of demand, and so forth. Hence the 2100 results tell us less about what to prepare for in the future, and more about how the model operates. The authors neglect to acknowledge this

modeling artefact, and instead present their 2100 results as a future of some likelihood and seeming confidence, which risks being misunderstood by less diligent readers.

Agreed. We have made edits to make our intention clearer.

See lines 142-147 (tack-changes version):

“We construct nine stock-driven scenarios to explore the evolution of cement-related materials until 2100 due to the longevity of buildings and infrastructures. Our scenarios are based on the mass balance principle, and assume that in-use cement stocks will eventually saturate in each of the 10 regions analyzed in different socioeconomic contexts (see Supplementary Fig. S1).”

Comment #9

8. Related to these issues, the authors present numbers with decimal-digit precision (e.g. 58.4%, 101.6 Gt...), where prudence would allow perhaps for rounding up to whole integers or even tens or hundreds (e.g. ~60%, ~100 Gt). This is relevant to all numbers in the text. Good luck!

We have revisited the numbers and made a few round-ups. We decided to not round up numbers that have to add up.

See edits throughout the manuscript.

Reviewers' comments:

Reviewer #2 (Remarks to the Author):

The paper has improved considerably and the authors have taken all my comments into account.

However, upon reading the feedback from other reviewers and the authors response to that, I'd like to second reviewer #3 that the range of scenarios presented in this study is extremely narrow from the perspective of long-term socioeconomic scenarios. Since the authors use a single population scenario in combination with a per-capita saturation level it matters a lot whether the global population will peak at 8 billion or grow to 12 billion. Moreover, the current method ignores GDP as a driving force across scenarios, which is fine for short term stock projects, but problematic when thinking about a world with 8 billion people at >150k USD/capita GDP vs 12 billion people at ~60k USD/capita GDP by the end of the century (see for instance the Shared Socioeconomic Pathways).

The response of the authors to the issues raised by Reviewer #3 is very unsatisfactory. They have basically added a reference to the UN population projection, without showing any acknowledgement of the fact that using a single population projection is a serious limitation for this study. They should either expand the discussion on this limitation, and clearly acknowledge that the 9 scenarios ignore socioeconomic variation, or add a scenario set with multiple population scenarios to the paper.

Reviewer #3 (Remarks to the Author):

The authors have clearly put effort into this revision, which is a great improvement over the previous submission. Many of the reviewers' comments have been thoughtfully acknowledged and dealt. The objective of this study is interesting and worthy of publication.

However, two of my major concerns have not been satisfactorily addressed. The authors' rebuttals to these issues are stylistic revisions that don't deal with the core of these issues, and additions of somewhat vague references to the SI. However, I don't think this kind of "tucking away" of core assumptions into the SI should be accepted. These issues are at the center of the research because they involve fundamental assumptions. Their effects propagate throughout the study and its take-away messages. and ignoring them impedes the authors' stated aims for this research. Dealing with these issues properly will get this research closer to publication in my opinion. Perhaps I haven't been clear enough about them before, so allow me to reiterate and rephrase:

1. Concealed growth-centered bias of your scenarios: All nine scenarios assume growth in per-capita cement stocks, the only variation is to the speed and final level of the stock. You don't have a single scenario that models constant or decreasing per-cap stocks, and so your scenarios are extremely restrictive. However, there is no reason to assume that only growth is an option. Cf. scenarios like SSP1 or the Low Energy Demand scenario <https://doi.org/10.1038/s41560-018-0172-6>

Because of the structure and balance of content between the paper and SI, this extremely important aspect is concealed. Assuming that most readers (including decision makers) don't delve into 100-page-long SI documents, the growth-bias of the scenarios and its scale is not communicated. The effect is a misleading message: there is no future of decreasing cement demand, and carbon emissions will continue to rise in the 'no action' case in all potential futures. This is improper, especially for a high-profile, broad-audience publication.

I think there are two options: (1) be outright about this bias in the main text and describe the research in the context of this bias, including in the abstract, methods, descriptions of the research objectives, and discussion of the biased results. Or (2) add scenarios of stable and decreasing per-cap stocks (perhaps instead of a few of the growth-biased scenarios).

2. Imposition of fast growth rates in the future: in several key regions (North America, Europe, Developed Asia & Oceania) your curve fitting choices lead to renewed accelerated growth of per-cap stocks in the future in regions where the past curves already exhibit nearly complete s-curve shapes. Their historical curves suggest that the s-curve's inflection point (point of fastest growth) has already occurred, yet the curve fitting choices impose a new inflection point occurring in the future. In the extreme case of the high-fast scenario, future annual growth rates near the inflection point are faster than any historical annual growth rates. I don't think this is reasonable. The authors don't give any rationalization for this (e.g. what kind of socio-economic mechanisms could lead per cap cement to suddenly jump in a very short period of 20 years from 2030-2050 in North America? What kind of world does this scenario describe?) As with the previous comment, not only do the authors not provide rationalizations, they do not even mention this peculiarity, and one must read the SI and interpret the figures to discover it. Either reconsider these curve fitting assumptions (enable the curve fitting algorithm to allow the inflection point to have occurred in the past, or flag your current extreme scenarios as "unfeasible") or be outright explicit about the results of the curve fits in the main text e.g. "Our assumptions lead in certain cases to unprecedented growth in per-cap cement stocks in developed regions, and the results should be interpreted in that context".

Two more important comments:

3. The aggregation of the regions to global-scale results may be a fallacy: The authors simply sum all of the regions within each scenario to present 9 global results. However, I suspect that if the historical global per-cap cement stock was fitted with s-curves, the resulting future stocks (and all other results that stem from this) would be quite different. Can you confirm this? If this is indeed the case, it has potentially serious implications because it may be the same if a region was disaggregated into countries, each with its own s-curve fit. This doesn't necessarily invalidate your results, but highlights the need to be very outright about the context for inference from your results.

4. I suspect that the uncertainties and sensitivities described in SI section 5.1 were estimated improperly, producing too-low uncertainties. One would assume that as time goes by, the uncertainties would increase over time, especially with a long time horizon until 2100. However, from the figures in SI section 5.1 and accompanying numbers in the excel SI it is clear that the uncertainty bands are functions of the value of Y (annual co2 emissions) and not of X (time). This creates an unreasonable result, in which the uncertainty in some intermediate years is higher than the uncertainty in 2100, clearly seen for example in R1_S8 and R1_S9 and many others.

A few more comments about clarity:

1. Line 52 "have likely since been sequestered" – by cement stocks or sequestered in general (by biomass, the oceans, etc.)? please clarify.
2. Lines 104-106: revise units from "10.2 t" to "10.2 t/cap".
3. Lines 121-122 sentence starting with "In addition" is unclear, both on its own and in relation to the rest of the paragraph. Please rephrase and clarify.
4. Line 130 "our scenarios are based on the mass balance principle" is an incorrect statement. Your models are based on the mass balance principle. The scenarios' premises have nothing to do with mass balance, they are based on your scenario assumptions.
5. Line 141: explain why the number 98% was chosen (i.e. why not 97% or 99%)?
6. Lines 154 and table 1: explicitly explain that E-M5 is CCS at the point of emission. This is very different from atmospheric CCS which is a measure you don't model (and rightly so).
7. Line 160 Table 1: explain why measure 4 is coded twice: E-M4 and U-M4. The text accompanying table 1 (lines 151-154) doesn't mention U-M4 at all; the mechanism is not explained in figure 2 (the dark green of U-M4 is not visible in the figures, only in the legend)
8. Line 171: define "phase displacement".
9. Line 174: this sentence is not backed by reference 33. First, reference 33 talks about steel stocks and here you talk about cement demand (inflows), so you make an unsubstantiated

comparison. Second, there are plenty of studies that are based on the opposite claim, that there are indeed long-term (dynamic) relations between inflows and economic activity. Citing this single study (whether relevant or not) seems like cherry-picking to me. Third, the IEA's numbers are scenarios, just like yours, and comparing their validity compared to your scenarios is counterproductive. Truth is, this entire sentence is unnecessary and should be removed.

10. Line 209 figure 2: as mentioned in an earlier comment, the dark-green of U-M4 is not visible. Furthermore, please explain what "U minus U-M4" means. Finally, since you're using the color green substantially in panel a in the 9 scenario figures, avoid using green in panel b S1-S3, to avoid suggesting a link between the color green in these two panels. Use yellow or some other unrelated color.

11. Lines 222 and 227: Mismatch between the text and figure 3: You refer to figure 3 when describing results in Gt, but figure 3 shows only % of the whole.

12. Figure 3: again, green appears in two different meanings: as "U" and as "0". Change one of them to a completely different color to avoid ambiguity.

13. Lines 253-255 are redundant. Recommend to remove them.

14. Line 291: define "asymptotically" or find a simpler way to explain your intention.

15. Figure 4 is unclear. In panel a, what do the negative temperature changes imply? Global cooling? And what is the gray shaded area in panel a, and how was it calculated? In panel b you compare the impulse response functions of the 10 regions but not of different time periods. Do these functions change over time? My understanding of your methodology is that they don't. And if they don't, then it doesn't matter that t_0 is 1930. You should just change the horizontal axis to "years since production" from 0 to 70.

16. Lines 334-346: add the codes from table 1 (E-M1 etc.) to ease comprehension.

17. SI lines 186-197 equations 3 and 4: I assume that like other dynamic MFA models, your model is discrete, not continuous. Therefore, change the integration notation to a sigma (cumulative sum) notation. If it's continuous, you need to explain the calculation mechanism.

SI line 187: Mismatch: "where $S_j(t_n - t_{n-1})$ " but in equation 3 it's $S_j(t, t_n)$ and t_{n-1} doesn't appear in equation 3 at all.

18. SI line 196-197 equation 4: why is the integral symbol's top interval t_{n-1} instead of n ? Please explain or revise.

19. SI line 201: standard deviation of 1/5 of the mean is quite steep. Have you tried other values, and how do they compare?

Good luck!

Response to reviewers for manuscript NCOMMS-19-14627A

“The sponge effect and carbon emission mitigation potentials of the global cement cycle”

Reviewer #2 (Remarks to the Author):

Comment #1

The paper has improved considerably and the authors have taken all my comments into account.

Thank you very much for your positive comments.

Comment #2

However, upon reading the feedback from other reviewers and the authors response to that, I'd like to second reviewer #3 that the range of scenarios presented in this study is extremely narrow from the perspective of long-term socioeconomic scenarios. Since the authors use a single population scenario in combination with a per-capita saturation level it matters a lot whether the global population will peak at 8 billion or grow to 12 billion. Moreover, the current method ignores GDP as a driving force across scenarios, which is fine for short term stock projects, but problematic when thinking about a world with 8 billion people at >150k USD/capita GDP vs 12 billion people at ~60k USD/capita GDP by the end of the century (see for instance the Shared Socioeconomic Pathways). The response of the authors to the issues raised by Reviewer #3 is very unsatisfactory. They have basically added a reference to the UN population projection, without showing any acknowledgement of the fact that using a single population projection is a serious limitation for this study. They should either expand the discussion on this limitation, and clearly acknowledge that the 9 scenarios ignore socioeconomic variation, or add a scenario set with multiple population scenarios to the paper.

Thanks for revisiting our assumptions and reviewer #3's assessment.

Following the suggestions from reviewer #2 and #3, we have revisited the messages delivered in our study, especially the main text. To make the context of our scenarios more explicit, we have revisited the introduction, methodology, and discussion.

See line 60-63:

“If the world follows a development pathway that is consistent with typical patterns observed in several industrialized countries, the global convergence of buildings and infrastructure services in all nations, to the level of these countries, is expected to drive sustained increases in global cement demand to build up the desired in-use stocks^{1,13-15}.”

See line 126-142:

“In light of the observed historical patterns of cement stocks and the essential role of in-use stock dynamics to the cement cycle, we simulate the future cement cycle in ten regions using a stock-driven approach¹⁶ based on the historical patterns of per capita cement stocks identified in our previous work¹ and a moderately growing population obtained from the medium scenario of

United Nations World Population Prospects³⁴. We deem the level of in-use cement stocks as an explicit physical representation of service provision to society, thereby constructing nine stock-driven scenarios to explore the evolution of cement-related materials until 2100 due to the longevity of buildings and infrastructures. Our scenarios assume that (i) per capita cement stocks in the ten regions follow a development path that is consistent with typical patterns (S-shaped curves) observed in the past century, but at different developmental stages¹; (ii) inequality of service provision delivered by cement stocks between industrialized and emerging regions will be reduced, and therefore, per capita cement stocks in the emerging regions will grow more rapidly (see Supplementary Fig. S1); (iii) the growth of per capita cement stocks in industrialized regions is still positive but slowing down, and therefore, the per capita cement stocks will continue growing but eventually saturate (see Supplementary Fig. S1); (iv) technological development for optimizing cement use in buildings and infrastructure proceeds, but without fundamental breakthroughs (e.g., new materials that replace cement to a full extent), because cement is a ubiquitous, relatively cheap building material of good workability.”

See line 272-277:

“Again, the narrative of our scenarios is essentially a development path where the growth of cement stocks does not shift markedly from historical patterns, implying that population and per capita cement stocks are fundamental drivers for the stock growth scenarios. Given this context, the varying saturation levels in our scenario analysis still highlight the urgent and precious opportunities to mitigate CO₂ emissions in emerging regions where buildings and infrastructures are yet to be constructed.”

See line 353-355:

“One of the fundamental assumptions in our scenarios is a moderately growing population, meaning that cement demand and demolition and associated CO₂ emissions and uptake would be significantly affected by population (see Supplementary Fig. S85-104).”

Also, we have conducted comprehensive sensitivity analyses on the population. We took three variants of population data (i.e., medium-variant, high-variant, and low-variant) from the United Nations population forecast.

See Figure S84-104.

Reviewer #3 (Remarks to the Author):

Comment #1

The authors have clearly put effort into this revision, which is a great improvement over the previous submission. Many of the reviewers’ comments have been thoughtfully acknowledged and dealt. The objective of this study is interesting and worthy of publication.

Again, we appreciate your positive comments and thorough assessment.

Comment #2

However, two of my major concerns have not been satisfactorily addressed. The authors' rebuttals to these issues are stylistic revisions that don't deal with the core of these issues, and additions of somewhat vague references to the SI. However, I don't think this kind of "tucking away" of core assumptions into the SI should be accepted. These issues are at the center of the research because they involve fundamental assumptions. Their effects propagate throughout the study and its take-away messages. and ignoring them impedes the authors' stated aims for this research. Dealing with these issues properly will get this research closer to publication in my opinion.

Perhaps I haven't been clear enough about them before, so allow me to reiterate and rephrase:

1. Concealed growth-centered bias of your scenarios: All nine scenarios assume growth in per-capita cement stocks, the only variation is to the speed and final level of the stock. You don't have a single scenario that models constant or decreasing per-cap stocks, and so your scenarios are extremely restrictive. However, there is no reason to assume that only growth is an option. Cf. scenarios like SSP1 or the Low Energy Demand scenario <https://doi.org/10.1038/s41560-018-0172-6>

Because of the structure and balance of content between the paper and SI, this extremely important aspect is concealed. Assuming that most readers (including decision makers) don't delve into 100-page-long SI documents, the growth-bias of the scenarios and its scale is not communicated. The effect is a misleading message: there is no future of decreasing cement demand, and carbon emissions will continue to rise in the 'no action' case in all potential futures. This is improper, especially for a high-profile, broad-audience publication.

I think there are two options: (1) be outright about this bias in the main text and describe the research in the context of this bias, including in the abstract, methods, descriptions of the research objectives, and discussion of the biased results. Or (2) add scenarios of stable and decreasing per-cap stocks (perhaps instead of a few of the growth-biased scenarios).

Thanks for pointing out these fundamental assumptions that are at the core of the narrative of our scenarios.

Following the reviewer's suggestion, we have clarified these assumptions in the main text, in case readers don't delve into supplementary information.

See line 60-63:

"If the world follows a development pathway that is consistent with typical patterns observed in several industrialized countries, the global convergence of buildings and infrastructure services in all nations, to the level of these countries, is expected to drive sustained increases in global cement demand to build up the desired in-use stocks^{1,13-15}."

See line 126-142:

“In light of the observed historical patterns of cement stocks and the essential role of in-use stock dynamics to the cement cycle, we simulate the future cement cycle in ten regions using a stock-driven approach¹⁶ based on the historical patterns of per capita cement stocks identified in our previous work¹ and a moderately growing population obtained from the medium scenario of United Nations World Population Prospects³⁴. We deem the level of in-use cement stocks as an explicit physical representation of service provision to society, thereby constructing nine stock-driven scenarios to explore the evolution of cement-related materials until 2100 due to the longevity of buildings and infrastructures. Our scenarios assume that (i) per capita cement stocks in the ten regions follow a development path that is consistent with typical patterns (S-shaped curves) observed in the past century, but at different developmental stages¹; (ii) inequality of service provision delivered by cement stocks between industrialized and emerging regions will be reduced, and therefore, per capita cement stocks in the emerging regions will grow more rapidly (see Supplementary Fig. S1); (iii) the growth of per capita cement stocks in industrialized regions is still positive but slowing down, and therefore, the per capita cement stocks will continue growing but eventually saturate (see Supplementary Fig. S1); (iv) technological development for optimizing cement use in buildings and infrastructure proceeds, but without fundamental breakthroughs (e.g., new materials that replace cement to a full extent), because cement is a ubiquitous, relatively cheap building material of good workability.”

See line 272-277:

“Again, the narrative of our scenarios is essentially a development path where the growth of cement stocks does not shift markedly from historical patterns, implying that population and per capita cement stocks are fundamental drivers for the stock growth scenarios. Given this context, the varying saturation levels in our scenario analysis still highlight the urgent and precious opportunities to mitigate CO₂ emissions in emerging regions where buildings and infrastructures are yet to be constructed.”

See line 353-355:

“One of the fundamental assumptions in our scenarios is a moderately growing population, meaning that cement demand and demolition and associated CO₂ emissions and uptake would be significantly affected by population (see Supplementary Fig. S85-104).”

As with sensitivity analyses in the 1st revision, we have conducted similar sensitivity analyses on the population. Please find details in Figure S85-104.

Comment #3

2. Imposition of fast growth rates in the future: in several key regions (North America, Europe, Developed Asia & Oceania) your curve fitting choices lead to renewed accelerated growth of per-cap stocks in the future in regions where the past curves already exhibit nearly complete s-curve shapes. Their historical curves suggest that the s-curve’s inflection point (point of fastest growth) has already occurred, yet the curve fitting choices impose a new inflection point occurring in the future. In the

extreme case of the high-fast scenario, future annual growth rates near the inflection point are faster than any historical annual growth rates. I don't think this is reasonable. The authors don't give any rationalization for this (e.g. what kind of socio-economic mechanisms could lead per cap cement to suddenly jump in a very short period of 20 years from 2030-2050 in North America? What kind of world does this scenario describe?) As with the previous comment, not only do the authors not provide rationalizations, they do not even mention this peculiarity, and one must read the SI and interpret the figures to discover it. Either reconsider these curve fitting assumptions (enable the curve fitting algorithm to allow the inflection point to have occurred in the past, or flag your current extreme scenarios as "unfeasible") or be outright explicit about the results of the curve fits in the main text e.g. "Our assumptions lead in certain cases to unprecedented growth in per-cap cement stocks in developed regions, and the results should be interpreted in that context".

We agree with your comments. As suggested, we now provide more rationalization on the narrative of our scenarios. Our assumption on industrialized regions is based on the observation in our previous study that the growth of per capita cement stocks in industrialized regions is still positive but slowing down. Therefore, we assume that the per capita cement stocks in these regions will continue growing but eventually saturate. Of course, the growth of the per capita cement stocks in these regions is relatively less significant.

See line 133-139:

"Our scenarios assume that (i) per capita cement stocks in the ten regions follow a development path that is consistent with typical patterns (S-shaped curves) observed in the past century, but at different developmental stages¹; (ii) inequality of service provision delivered by cement stocks between industrialized and emerging regions will be reduced, and therefore, per capita cement stocks in the emerging regions will grow more rapidly (see Supplementary Fig. S1); (iii) the growth of per capita cement stocks in industrialized regions is still positive but slowing down, and therefore, the per capita cement stocks will continue growing but eventually saturate (see Supplementary Fig. S1);

The explanation for envisaging a few extreme scenarios in a few regions (i.e., North America, Europe, and Developed Asia & Oceania) is that our scenarios are based on a moderately growing population and per capita cement stocks in the world follow a development path. According to the newly added sensitivity analyses on population (see more results in Figure S85), if the population take the low-variant path, the cement inflows will level off and decrease eventually.

Comment #4

Two more important comments:

3. The aggregation of the regions to global-scale results may be a fallacy: The authors simply sum all of the regions within each scenario to present 9 global results. However, I suspect that if the historical global per-cap cement stock was fitted with s-curves, the resulting future stocks (and all other results that stem from this) would be quite different. Can you confirm this? If this is indeed the case, it has potentially serious implications because it may be the same if a region was disaggregated into countries, each with its own s-curve fit. This doesn't necessarily invalidate your results, but highlights the need to be very outright about the context for inference from your results.

Point taken. We have added a new calculation and a few figures in Supplementary Source Data - Per_Cap_Stock_World. Indeed, what you suggested should be more explicitly acknowledged.

Following your suggestion, we have added more explanation to make this issue explicit. In addition, we also added a few lines in Section - Limitations and uncertainty to elaborate our uncertainty analysis.

See line 340-347:

“Country-specific modeling of the cement cycle requires country-specific assumptions on future stock development, whereas global modeling could not reflect the discrepancies between industrialized and emerging regions. Besides, pairing the country-specific cement cycle with the other two layers requires relevant country-specific understanding. As a compromise, 184 countries are aggregated into ten regions (i.e., North America, Latin America & Caribbean, Europe, Commonwealth of Independent States, Africa, Middle East, India, China, Developed Asia & Oceania, and Developing Asia), each comprising countries with similar socioeconomic and geographic circumstances.”

See line 416-428:

“Although it differentiates the discrepancies among different regions, the global ten-region model can be further improved if country-specific assumptions are available. Beyond this, the

main sources of uncertainty are firstly in the global stock-flow model, and secondly in the cement carbonation model. The first is mainly accounted for through the range of saturation times and levels in the nine scenarios. The effect of different population forecasts is also explored through a sensitivity analysis (see Supplementary Fig. S85-104). For the second set of uncertainties about the cement carbonation effect, we employed the same Monte Carlo method and parameters used in the global cement carbonation model⁵ to estimate uncertainties in CO₂ uptake. Critical causes of uncertainties associated with carbonation were identified and their impacts on simulation results were evaluated by the Monte Carlo method recommended by the 2006 IPCC guidelines for National Greenhouse Gas Inventories⁵³ (see Supplementary Fig. S54-63). Likewise, we employed the Monte Carlo method to estimate uncertainties in CO₂ emissions following the practice recommended by the 2006 IPCC guidelines⁵³ (see Supplementary Fig. S44-53).”

Comment #5

4. I suspect that the uncertainties and sensitivities described in SI section 5.1 were estimated improperly, producing too-low uncertainties. One would assume that as time goes by, the uncertainties would increase over time, especially with a long time horizon until 2100. However, from the figures in SI section 5.1 and accompanying numbers in the excel SI it is clear that the uncertainty bands are functions of the value of Y (annual co2 emissions) and not of X (time). This creates an unreasonable result, in which the uncertainty in some intermediate years is higher than the uncertainty in 2100, clearly seen for example in R1_S8 and R1_S9 and many others.

We agree with you that uncertainties would increase over time. This type of uncertainty is dealt with the sensitivity analyses in our study. The Monte Carlo method used in our study looks at the inherent uncertainties (e.g., statistical variation) of the parameters of our model.

Besides, as detailed in the passages before SI Section 3.1, we used a Normal distribution with a relative standard deviation (%) for different parameters. We have added a new calculation on the **relative** uncertainties (95% confidence intervals) in Supplementary Source Data - Fig.S44-53, showing that the **relative** uncertainties keep nearly constant.

Comment #6

A few more comments about clarity:

1. Line 52 “have likely since been sequestered” – by cement stocks or sequestered in general (by biomass, the oceans, etc.)? please clarify.

We have corrected it.

See line 52-53:

“.....have likely since been sequestered by cement-related materials.....”

Comment #7

2. Lines 104-106: revise units from “10.2 t” to “10.2 t/cap”.

We have corrected it.

See line 107:

“.....average cement stocks per capita reached 10.2 t/cap in 2014.....”

Comment #8

3. Lines 121-122 sentence starting with “In addition” is unclear, both on its own and in relation to the rest of the paragraph. Please rephrase and clarify.

Point taken. We have moved to this sentence to the next paragraph to make the transition smoother.

See line 130-131:

“We deem the level of in-use cement stocks as an explicit physical representation of service provision to society.....”

Comment #9

4. Line 130 “our scenarios are based on the mass balance principle” is an incorrect statement. Your models are based on the mass balance principle. The scenarios’ premises have nothing to do with mass balance, they are based on your scenario assumptions.

We agree with your comment. We have rephrased this sentence and added a few lines to rationalize our assumptions.

See line 133-142:

“Our scenarios assume that (i) per capita cement stocks in the ten regions follow a development path that is consistent with typical patterns (S-shaped curves) observed in the past century, but at different developmental stages¹; (ii) inequality of service provision delivered by cement stocks between industrialized and emerging regions will be reduced, and therefore, per capita cement stocks in the emerging regions will grow more rapidly (see Supplementary Fig. S1); (iii) the growth of per capita cement stocks in industrialized regions is still positive but slowing down, and therefore, the per capita cement stocks will continue growing but eventually saturate (see Supplementary Fig. S1); (iv) technological development for optimizing cement use in buildings and infrastructure proceeds, but without fundamental breakthroughs (e.g., new materials that replace cement to a full extent), because cement is a ubiquitous, relatively cheap building material of good workability.”

Comment #10

5. Line 141: explain why the number 98% was chosen (i.e. why not 97% or 99%)?

The number 98% is an arbitrary choice as a means to parameterize the speed of the growth curve, giving an intuitive timescale by which most of the growth has occurred. We have rephrased this sentence to make it more explicit.

See line 152-154:

“The saturation time reflects the speed of stock growth (parameterized by the time when the per capita cement stocks reach 98% of the saturation level).”

Comment #11

6. Lines 154 and table 1: explicitly explain that E-M5 is CCS at the point of emission. This is very different from atmospheric CCS which is a measure you don't model (and rightly so).

Thanks for your suggestion. We have added a few words to clarify this.

See Table 1:

“25% of CO₂ emissions from cement production captured in cement plants by 2050”

Comment #12

7. Line 160 Table 1: explain why measure 4 is coded twice: E-M4 and U-M4. The text accompanying table 1 (lines 151-154) doesn't mention U-M4 at all; the mechanism is not explained in figure 2 (the dark green of U-M4 is not visible in the figures, only in the legend)

Thanks for your suggestion.

We have added some explanation in the legend of Table 1 as well as the legend of Figure 2.

Comment #13

8. Line 171: define “phase displacement”.

We have rephrased it and added a few more words to make it more understandable.

See line 182-185:

“The gradual rise and then saturation of in-use stocks lead to cyclical variations in global cement demand over the next decades (see Supplementary Fig. S22), while the global demolition waste generation continues to rise due to the delay between demand and demolition caused by the longevity of in-use cement stocks (see Supplementary Fig. S33)”

Comment #14

9. Line 174: this sentence is not backed by reference 33. First, reference 33 talks about steel stocks and here you talk about cement demand (inflows), so you make an unsubstantiated comparison. Second, there are plenty of studies that are based on the opposite claim, that there are indeed long-term (dynamic) relations between inflows and economic activity. Citing this single study (whether relevant or not) seems like cherry-picking to me. Third, the IEA's numbers are scenarios, just like yours, and comparing their validity compared to your scenarios is counterproductive. Truth is, this entire sentence is unnecessary and should be removed.

We agree with your comment. Indeed, reference 33 is focused on steel stocks and flows. After revisiting this paragraph, we agree with you that this sentence is an overstatement, and therefore decided to delete this sentence.

Comment #15

10. Line 209 figure 2: as mentioned in an earlier comment, the dark-green of U-M4 is not visible. Furthermore, please explain what “U minus U-M4” means. Finally, since you’re using the color green substantially in panel a in the 9 scenario figures, avoid using green in panel b S1-S3, to avoid suggesting a link between the color green in these two panels. Use yellow or some other unrelated color.

Thanks for pointing out this coloring issue. We have changed the color of U to avoid unnecessary associations between Figure 2a and 2b.

Besides, we have added some explanation for U-M4, which is consistent with the explanation in the legend of Table 1.

See line 232-233:

“U-M4: clinker substitution marginally reduces CO₂ uptake in cement related materials.”

a

b

Comment #16

11. Lines 222 and 227: Mismatch between the text and figure 3: You refer to figure 3 when describing results in Gt, but figure 3 shows only % of the whole.

We would like to reiterate that, in Figure 3, the red-yellow-green gradient presents the absolute amount of ‘no-action’ CO₂ emissions in each region.

The column graphs below the red-yellow-green gradient present the relative contribution (%).

Comment #17

12. Figure 3: again, green appears in two different meanings: as “U” and as “0”. Change one of them to a completely different color to avoid ambiguity.

Point taken. We have replotted Figure 3.

Comment #18

13. Lines 253-255 are redundant. Recommend to remove them.

Point taken. We have deleted this sentence.

Comment #19

14. Line 291: define “asymptotically” or find a simpler way to explain your intention.

We have rephrased this sentence.

See line 307-308:

“.....(i.e., CO₂ emissions from biomass combustion are gradually re-captured by biomass regrowth⁴³ and thus net emissions tend to zero).”

Comment #20

15. Figure 4 is unclear. In panel a, what do the negative temperature changes imply? Global cooling? And what is the gray shaded area in panel a, and how was it calculated? In panel b you compare the impulse response functions of the 10 regions but not of different time periods. Do these functions change over time? My understanding of your methodology is that they don't. And if they don't, then it doesn't matter that t_0 is 1930. You should just change the horizontal axis to "years since production" from 0 to 70.

The negative temperature change presents the preventable warming relative to the period 2010-2019. We have added "preventable" to make it clear.

For Figure 4b, the time does matter because CO₂ emissions of one-tonne cement change over time due to improvements in cement production. One can imagine that each line moves down with the same line shape if CO₂ emissions of one-tonne cement are reduced over time.

Comment #21

16. Lines 334-346: add the codes from table 1 (E-M1 etc.) to ease comprehension.

We have added the codes to improve the comprehensibility of methods.

Comment #22

17. SI lines 186-197 equations 3 and 4: I assume that like other dynamic MFA models, your model is discrete, not continuous. Therefore, change the integration notation to a sigma (cumulative sum) notation. If it's continuous, you need to explain the calculation mechanism.

Thanks so much for reading through the SI. Indeed, our model is discrete. Following the reviewer's suggestion, we have replaced the integral symbol with the sum symbol.

Comment #23

SI line 187: Mismatch: "where $S_j(t_n - t_{n-1})$ " but in equation 3 it's $S_j(t, t_n)$ and t_{n-1} doesn't appear in equation 3 at all.

We have corrected this typo.

Comment #24

18. SI line 196-197 equation 4: why is the integral symbol's top interval t_{n-1} instead of n ? Please explain or revise.

We have corrected this typo.

Comment #25

19. SI line 201: standard deviation of 1/5 of the mean is quite steep. Have you tried other values, and how do they compare?

We have compared this with some literature and standard deviation of the lifetime used in the previous DMFA studies, e.g., on steel and aluminum, which is higher (30%).

See SI line 209-214:

“The standard deviation of the normal distribution is set as 1/5 of the mean, which is smaller than those employed in the previous DMFA studies on steel and aluminum^{2,3}. This is because cement is ubiquitously used in buildings and structures of which the demolition tends to concentrate around the mean of the lifetime. Unlike cement, steel and aluminum are used in products of which the discarding tends to happen more evenly. One can expect that if the standard deviation is greater or smaller, the curve of lifetime distribution will have a longer or shorter tail.”

To help understand the effect of the standard deviation, we have plotted a new graph (see Supplementary Source Data - Lifetime), as well as a few new sentences in the Supplementary Information.

Reviewers' comments:

Reviewer #2 (Remarks to the Author):

The authors have clearly put great effort in revising the manuscript to the last review comments. The text is now much more explicit about the hidden limitations and assumptions in the scenarios and the authors have added a sensitivity analysis on different population projections in the SI.

My main concerns have been dealt with in a satisfactory way.

Reviewer #3 (Remarks to the Author):

The authors have not responded in a satisfactory manner to the reviewers' comments. Although they added some text to rationalize their scenario choices, this added text is not enough to disclose the growth-bias that the authors introduce in their scenarios, and not enough to disclose that there are, in fact, an arbitrarily large number of alternative scenarios that don't assume continued per-capita growth in all regions.

Unfortunately, I'm afraid that the issue is not the choice of population growth trends, but rather it's the per-capita cement values in the future. I regret having to repeat myself, but as I wrote in the previous two rounds, the authors chose 9 scenarios in which even developed regions will undergo growth in per-capita cement, and in 6/9 of the scenarios this is a substantial increase compared to current values. Furthermore, even though the authors repetitively write that "saturation is evident in several highly-developed countries" (lines 117-118) and "the growth of per capita cement stocks in industrialized regions is still positive but slowing down, and therefore, the per capita cement stocks will continue growing but eventually saturate" (lines 137-139), that's not what their scenarios model. According to the authors' own calculations, no region in no sector reached 10t/cap by 2015. The highest is nearly 9 t/cap. However, as seen in figure S1, table S3, and figures S2-S11, the scenarios assume that future per-capita cement will exceed these values in nearly all cases. Furthermore, I asked before, but didn't get answers to the peculiar shapes of future growth: for example in the NA and DOA regions, whose historical data already shows a slowdown towards saturation, why do all scenarios describe a renewed, faster, growth? These points conflict with the authors' statements, such as that "the narrative of our scenarios is essentially a development path where the growth of cement stocks does not shift markedly from historical patterns" lines 273-274.

The extremely scenario choices for China, which force its per-capita cement stocks to saturate at levels higher than current developed regions even in the low scenarios, and which force very rapid growth in all scenarios, coupled with its high population, act as a major driver of the results, but this is not acknowledged.

The outcome of all of this is that the authors "conclude that deep decarbonization of the global cement cycle calls for both 'passive CO2 sequestration' and 'active CO2 sequestration', but also that these measures are likely not enough to reach the 1.5°C climate goal – more innovative or drastic approaches are needed." (lines 219-222) – although perhaps what's needed is to limit per-capita stock levels to less than 10 t/cap as seen in developed countries today. Since despite a plethora of scenarios, such a scenario is not explored, of course the authors can't propose it as an option.

This is extremely disappointing, because all in all, the technical work of this study is remarkable – I find the research question interesting, and the methods, data processing, and calculations thought out and conducted well. I don't question any of those aspects. However the scenario choices and their interpretations remain flawed despite the repeated rounds of revisions, and hence cloud over the rest of the research.

Lastly, I was surprised to see a paper recently published by several of this study's authors on the same topic of carbon sequestration in cement:

<https://www.sciencedirect.com/science/article/pii/S1364032119307038> which was not disclosed to the reviewers during the review processes, and is not mentioned or cited in this study. Although

the two studies have sufficient technical differences in their objectives and methods and so are definitely not plagiarizing each other, the similarities are obvious and high. Since the reviewers are also asked to judge the novelty of this study, I believe the existence of another paper under review should have been disclosed to support the reviewers' task. The reviewers have questioned the novelty of this study already in the first round of reviews. In conclusion, I regretfully conclude that this study doesn't reach Nature Communications' expected level of novelty, impact, and outstanding research for publication. I dishearteningly recommend a rejection.

Reply to comments from Reviewer #2:

Comment #1

The authors have clearly put great effort in revising the manuscript to the last review comments. The text is now much more explicit about the hidden limitations and assumptions in the scenarios and the authors have added a sensitivity analysis on different population projections in the SI.

My main concerns have been dealt with in a satisfactory way.

Thank you so much for your comments and we appreciate your time and acknowledgement for our work.

Reply to comments from reviewer #3:

Comment #1

The authors have not responded in a satisfactory manner to the reviewers' comments. Although they added some text to rationalize their scenario choices, this added text is not enough to disclose the growth-bias that the authors introduce in their scenarios, and not enough to disclose that there are, in fact, an arbitrarily large number of alternative scenarios that don't assume continued per-capita growth in all regions.

Unfortunately, I'm afraid that the issue is not the choice of population growth trends, but rather it's the per-capita cement values in the future. I regret having to repeat myself, but as I wrote in the previous two rounds, the authors chose 9 scenarios in which even developed regions will undergo growth in per-capita cement, and in 6/9 of the scenarios this is a substantial increase compared to current values. Furthermore, even though the authors repetitively write that "saturation is evident in several highly-developed countries" (lines 117-118) and "the growth of per capita cement stocks in industrialized regions is still positive but slowing down, and therefore, the per capita cement stocks will continue growing but eventually saturate" (lines 137-139), that's not what their scenarios model. According to the authors' own calculations, no region in no sector reached 10t/cap by 2015. The highest is nearly 9 t/cap. However, as seen in figure S1, table S3, and figures S2-S11, the scenarios assume that future per-capita cement will exceed these values in nearly all cases. Furthermore, I asked before, but didn't get answers to the peculiar shapes of future growth: for example in the NA and DOA regions, whose historical data already shows a slowdown towards saturation, why do all scenarios describe a renewed, faster, growth? These points conflict with the authors' statements, such as that "the narrative of our scenarios is essentially a development path where the growth of cement stocks does not shift markedly from historical patterns" lines 273-274.

Thank you so much for your time and comments.

We have revisited our scenario storylines. We highly appreciate the reviewer's comments on having alternative scenarios that do not assume continued per-capita growth in all regions. Upon careful internal discussion, we have redesigned our scenario narratives and parameters, inspired by the Resource Efficiency-Climate Change Nexus (RECC) scenario modeling framework (for example, as recently developed by Hertwich, Fishman, and colleagues, e.g., Fishman, T. et al. Developing scenarios of resource efficiency and climate change: from conception to operation. <https://osf.io/tqsc3> (2020) doi:10.31235/osf.io/tqsc3.) Similarly, we envisage three scenario storylines with varying levels of cement stocks, with the first scenario storyline (S1-3) characterized by a low cement stock level, the second scenario storyline (S4-6) by a medium cement stock level, and the third scenario storyline (S7-9) by a high cement stock level. The saturation level of per capita cement stocks is regarded as a tangible indicator for various human needs in mature societies, including shelter, transport networks, factories, offices, as well as commercial, educational, healthcare, and governmental facilities. It is the level of *service* provided by per capita cement stocks that are expected to saturate, not just the quantity of *material* involved; the two are linked by the material intensity of the in-use product stocks.

Feeding with the newly formulated scenarios, we have performed a new round of simulations.

All figures in the main text and SI have been updated accordingly. The main text has been edited as well.

Comment #2

The extremely scenario choices for China, which force its per-capita cement stocks to saturate at levels higher than current developed regions even in the low scenarios, and which force very rapid growth in all scenarios, coupled with its high population, act as a major driver of the results, but this is not acknowledged.

We have rephrased our scenario narratives to include an acknowledgment of the underlying assumptions for China's scenario choices. Assumption #2 is specific for this comment.

See line 148-158:

“Our scenarios assume that (i) per capita cement stocks in the ten regions follow a development path that is consistent with S-shaped curves or inverted S-shaped curves toward a global convergence of per capita cement stocks, and therefore, regions or end-use sectors that have a per capita cement stock below the saturation level will see a continuing growth, while those with a per capita cement stock over the saturation level will see a decline (see Supplementary Fig. S1); (ii) the formulated pathways of per capita cement stocks do not entail abrupt changes in resulting cement demand, and therefore, the development pathways of per capita cement stocks in a few regions or end-use sectors are adjusted to smoothen the trends in cement demand; (iii) technological development for optimizing cement use in buildings and infrastructure proceeds, but without fundamental breakthroughs (e.g., new materials that replace cement to a full extent), because cement is a ubiquitous, relatively cheap building material of good workability.”

Comment #3

The outcome of all of this is that the authors “conclude that deep decarbonization of the global cement cycle calls for both ‘passive CO₂ sequestration’ and ‘active CO₂ sequestration’, but also that these measures are likely not enough to reach the 1.5°C climate goal – more innovative or drastic approaches are needed.” (lines 219-222) – although perhaps what's needed is to limit per-capita stock levels to less than 10 t/cap as seen in developed

countries today. Since despite a plethora of scenarios, such a scenario is not explored, of course the authors can't propose it as an option.

This is extremely disappointing, because all in all, the technical work of this study is remarkable – I find the research question interesting, and the methods, data processing, and calculations thought out and conducted well. I don't question any of those aspects. However the scenario choices and their interpretations remain flawed despite the repeated rounds of revisions, and hence cloud over the rest of the research.

Per the reviewer's suggestion, we have completely changed our scenario storylines by limiting per-capita stock levels to less than 10 t/cap. Storyline-consistent target values of per capita cement stocks are all less than 10 t/cap. Plus, we have articulated a few key assumptions for our scenario parameterization.

- (i) per capita cement stocks in the ten regions follow a development path that is consistent with S-shaped curves or inverted S-shaped curves toward a global convergence of per capita cement stocks, and therefore, regions or end-use sectors that have a per capita cement stock below the saturation level will see a continuing growth, while those with a per capita cement stock over the saturation level will see a decline (see Supplementary Fig. S1);
- (ii) the formulated pathways of per capita cement stocks do not entail abrupt changes in resulting cement demand, and therefore, the development pathways of per capita cement stocks in a few regions or end-use sectors are adjusted to smoothen the trends in cement demand;
- (iii) technological development for optimizing cement use in buildings and infrastructure proceeds, but without fundamental breakthroughs (e.g., new materials that replace cement to a full extent), because cement is a ubiquitous, relatively cheap building material of good workability.

The first assumption explains why some regions or sectors, who have a cement stock level above the target value, are declining.

The second assumption explains why some regions or sectors have to plateau first and then start to decline.

The third assumption explains why per capita cement stocks will still maintain a considerably high level.

With the newly formulated scenarios, we have been able to assess the plausibility of meeting climate change goals by exploring a low per capita cement stock level. As Fig.2a reveals, CO₂ balance (black lines) will maintain an overall stable trend in low cement stock scenarios (S1-S3). Meanwhile, net CO₂ balance (brown lines) in low cement stock scenarios (S1-S3) will decline more rapidly compared to S4-S9; however, they are still not low enough to meet the '1.5 °C limit'. Therefore, our previous conclusions still hold, albeit requiring changes to associated numbers in the texts. We have revised them throughout our manuscript.

Besides, we have also included some text in our discussion to articulate that the analytical results presented in this study should always be interpreted within the formulated scenario narratives (see line 283-284).

Comment #4

Lastly, I was surprised to see a paper recently published by several of this study's authors on the same topic of carbon sequestration in cement: <https://www.sciencedirect.com/science/article/pii/S1364032119307038> which

was not disclosed to the reviewers during the review processes, and is not mentioned or cited in this study. Although the two studies have sufficient technical differences in their objectives and methods and so are definitely not plagiarizing each other, the similarities are obvious and high. Since the reviewers are also asked to judge the novelty of this study, I believe the existence of another paper under review should have been disclosed to support the reviewers' task. The reviewers have questioned the novelty of this study already in the first round of reviews. In conclusion, I regretfully conclude that this study doesn't reach Nature Communications' expected level of novelty, impact, and outstanding research for publication. I dishearteningly recommend a rejection.

Thanks for raising this issue.

This is a article that is part of our long term efforts on understanding the role of built environment stocks and construction materials in climate change mitigation. Indeed, this study also looks at carbon sequestration in cement-related materials. However, the scope of the published study is foudamentally different from our manuscript here and limited to the carbon absorption potential of concrete debris generated at *the demolition stage*. Besides, this study does not consider the dynamics of cement outflows, nor the mass-balance between cement inflows and cement outflows. As the reviewer suggests, the two studies have sufficient technical differences in their objective/scope and methods. That being said, this study does not impair the novelty of our work (the published one addresses an issue that has been mentioned in the literature already and complemented with an updated estimation of global potentials, while this manuscript goes much beyond with integration of global cement cycle and demand forecasting, emissions accounting of the global cement cycle, and carbonation potential, both retrospectively and prospectively).

We have acknowledged this study by citing it in the introduction. See line 50-52:

“The scale of historical CO₂ absorption occurred along the entire cement cycle, or during certain life cycle stages, has been estimated regionally^{6,8} and globally^{5,9}”

REVIEWERS' COMMENTS:

Reviewer #2 (Remarks to the Author):

I've been asked to provide another review of this paper, even though I already had recommended its acceptance, because Reviewer #3 is not available. My original comments aligned very much with Reviewer #3's comments, and I realized that I agreed with most of what Reviewer #3 wrote during the last round.

Regarding the scenario projections, I think that Reviewer #3 had a valid point and the authors responded well by revising their scenario approach. The new scenarios are much more in line with the description and historic trends.

On the 1.5 degree scenario, the authors have improved the methods by making use of their new scenario logic and created a scenario that can be defended as a low-intensity 1.5 degree scenario.

Regarding comment #4 on the other paper, I agree with Reviewer #3 that this paper should have been mentioned or provided in the review process and that the novelty of this paper is impeded by the paper in Renewable and Sustainable Energy Reviews. The difference between the papers as the authors describe it is rather artificial. However, many of the main points of the current paper are still novel here as the other paper solely focuses on the waste flow. Therefore, I would not go as far as Reviewer #3 in rejecting this paper because of the other publication. In my view, the current paper is still a valuable contribution to the literature.

Response to reviewers comments on manuscript NCOMMS-19-14627C

“The sponge effect and carbon emission mitigation potentials of the global cement cycle”

Reviewer #2 (Remarks to the Author):

Comment #1

I've been asked to provide another review of this paper, even though I already had recommended its acceptance, because Reviewer #3 is not available. My original comments aligned very much with Reviewer #3's comments, and I realized that I agreed with most of what Reviewer #3 wrote during the last round.

We are grateful for your comments. Thank you so much for taking the time to review our 3rd revision.

Comment #2

Regarding the scenario projections, I think that Reviewer #3 had a valid point and the authors responded well by revising their scenario approach. The new scenarios are much more in line with the description and historic trends. On the 1.5 degree scenario, the authors have improved the methods by making use of their new scenario logic and created a scenario that can be defended as a low-intensity 1.5 degree scenario.

Thanks. Indeed, we have revisited our storylines, taking into account Review #3's comments. By doing so, we have been able to improve the validity and spectrum of our storylines.

The new cement stock scenarios have enabled us to explore a wide range of decarbonization possibilities.

Comment #3

Regarding comment #4 on the other paper, I agree with Reviewer #3 that this paper should have been mentioned or provided in the review process and that the novelty of this paper is impeded by the paper in Renewable and Sustainable Energy Reviews. The difference between the papers as the authors describe it is rather artificial. However, many of the main points of the current paper are still novel here as the other paper solely focuses on the waste flow. Therefore, I would not go as far as Reviewer #3 in rejecting this paper because of the other publication. In my view, the current paper is still a valuable contribution to the literature.

Thank you so much for your comments. We have made some edits in the introduction to articulate the limitations of the RSER study. We believe that having this point articulated will justify the novelty of our study.

“Understanding the mitigation potential of the sponge effect requires looking to the future, but future scenarios are often either based on cement demand linked to market growth^{9,10} or economic indicators^{11,12}, or limited to a certain life cycle stage (e.g., end-of-life demolition waste¹³). A proper, holistic understanding of the sponge effect requires not just forecasting cement demand but also a physically-consistent accounting of the cement stocks in the built environment and end-of-life demolition waste, where the carbonation actually occurs, and the cement demand for replacement and expansion of stocks.”